# Frequent transitions in self-assembly across the evolution of a central metabolic enzyme

Franziska L. Sendker [1,8], Tabea Schlotthauer[1,8], Christopher-Nils Mais[2], Yat Kei Lo[2], Mathias Girbig [1], Stefan Bohn[3], Thomas Heimerl[2], Daniel Schindler [2,4], Arielle Weinstein[5], Brian P. H. Metzger [5], Joseph W. Thornton [5,6], Arvind Pillai[5], Gert Bange [1,2,7], Jan M. Schuller [2,7] & Georg K. A. Hochberg [1,2,7] ✉

Many enzymes assemble into homomeric protein complexes comprising multiple copies of one protein. Because structural form is usually assumed to follow function in biochemistry, these assemblies are thought to evolve because they provide some functional advantage. In many cases, however, no specific advantage is known and, in some cases, quaternary structure varies among orthologs. This has led to the proposition that self-assembly may instead vary neutrally within protein families. The extent of such variation has been difficult to ascertain because quaternary structure has until recently been difficult to measure on large scales. Here, we employ mass photometry, phylogenetics, and structural biology to interrogate the evolution of homo-oligomeric assembly across the entire phylogeny of prokaryotic citrate synthases – an enzyme with a highly conserved function. We discover a menagerie of different assembly types that come and go over the course of evolution, including cases of parallel evolution and reversions from complex to simple assemblies. Functional experiments in vitro and in vivo indicate that evolutionary transitions between different assemblies do not strongly influence enzyme catalysis. Our work suggests that enzymes can wander relatively freely through a large space of possible assembly states and demonstrates the power of characterizing structure-function relationships across entire phylogenies.

Proteins commonly fulfill their physiological functions not as an individual polypeptide chain but as multimeric complexes formed by two or more copies of the same protomer that assemble via non-covalent interactions[1–3]. Self-assembly into homo-oligomers requires specific interfaces and complementary interactions between many amino acids. Because these features seem unlikely to originate just by chance, self-assembly is often assumed to be functionally advantageous[4,5].

Proposed advantages of self-assembly include efficient encoding and easier folding of large structures, increasing productive encounters with substrate, reducing vulnerability to degradation, and particular forms of allosteric regulation[1]. But cases in which a functional benefit has been clearly demonstrated for homo-oligomeric assembly[6–8] are almost certainly outnumbered by those where none is known. Self-assembly can also vary between orthologs that at least in theory fulfill

[1]Max-Planck-Institute for Terrestrial Microbiology, Karl-von-Frisch-Str. 10, 35043 Marburg, Germany. [2]Center for Synthetic Microbiology (SYNMIKRO), Philipps-University Marburg, Karl-von-Frisch-Str. 14, 35043 Marburg, Germany. [3]Helmholtz Munich Cryo-Electron Microscopy Platform, Helmholtz Munich, Ingolstädter Landstraße 1, Neuherberg, Germany. [4]MaxGENESYS Biofoundry, Max-Planck-Institute for Terrestrial Microbiology; Karl-von-Frisch-Str. 10, 35043 Marburg, Germany. [5]Department of Ecology and Evolution, University of Chicago, Chicago, IL, USA. [6]Department of Human Genetics, University of Chicago, Chicago, IL, USA. [7]Department of Chemistry, Philipps-University Marburg; Hans-Meerwein-Str. 4, 35043 Marburg, Germany. [8]These authors contributed equally: Franziska L. Sendker, Tabea Schlotthauer. ✉e-mail: georg.hochberg@mpi-marburg.mpg.de

the same function[9–11], leading to the proposition that proteins could drift randomly in and out of different, but equally functional forms of self-assembly[12].

The scale of variation in self-assembly within protein families is not well understood, because until recently self-assembly state was difficult to measure. In most cases, our knowledge derives from X-ray or cryo-electron microscopy structures that are usually only available for at best a handful of orthologs from often closely related species and also do not necessarily represent the physiological form of the protein due to e.g. crystallization artefacts[13]. Techniques to measure self-assembly states in solution like small-angle X-ray scattering or analytical size exclusion chromatography have relatively low resolution and can struggle to separate interconverting assemblies[14]. High-resolution native mass spectrometry is still low throughput and has very specific buffer requirements[15]. As a consequence, we have little data on the extent of variation in self-assembly state and its connection to biochemical function across whole protein families. Two recent studies retraced the evolution of self-assembly across the phylogeny of geranylgeranylglyceryl phosphate synthase[16] and a subfamily of ribulose-1,5-bisphosphate carboxylase/oxygenase (Rubisco)[17]. In both cases, novel stoichiometries first evolved without a major impact on function, and were subsequently lost several times, consistent with a largely neutral acquisition and loss of different types of self-assembly.

Here we extend this approach and trace the evolution of self-assembly across the phylogenetic history of an entire protein family using mass photometry – a fast and single-particle based technique to quantify protein assemblies[18] at physiologically relevant, nanomolar concentrations. We discover a whole range of previously unknown assemblies that interconvert frequently, including enzymes that populate several asymmetrical assembly states concurrently. We solved the structure of two previously unknown stoichiometries to reveal the biochemical mechanisms by which evolutionary transitions between oligomeric states occur. Our results add to a growing set of observations that self-assembly into higher-order oligomers is a remarkably plastic trait, even in proteins with highly conserved functions.

## Results

### Phylogenetic classification of citrate synthases

We chose to investigate the diversity in self-assembly in bacterial citrate synthases (CS), because the enzyme has a highly conserved function: CS catalyzes the first step of the citric acid cycle. Previous studies identified two different structural types of CS: type I dimers[19–21] and type II hexamers[22–24]. The type II CS from *E. coli* is known to be allosterically regulated by NADH via a mechanism that relies on hexamerization[25]. Other hexameric type II CS are not affected by NADH[23,24,26,27]. These observations imply at least one transition in homo-oligomeric assembly plus a regulatory novelty associated with assembly, making CS an appropriate protein family to study conservation and functional relevance of quaternary structure evolution.

We inferred a comprehensive phylogenetic tree of bacterial CS enzymes (Fig. 1a, Supplementary Fig. 1). Rooting of bacterial phylogenies is difficult because of the number of horizontal transfer events and there being no consensus on overall phylogenetic relationships of bacterial phyla[28,29]. It was also not possible to use Archaea for outgroup-rooting because the archaeal CS did not form a monophyletic group so we decided not to root the CS phylogeny. A parsimonious inference would locate the position of the root somewhere within type I CS, which contain all archaeal CS sequences as well as CS from several major bacterial phyla considered to differentiate early in different phylogenetic analyses[30–32]. Consistent with previous studies[27,33,34] our tree clearly separates type I and type II CS (Fig. 1a) but overall indicates a great amount of horizontal transfer events between bacteria, archaea and even eukaryotes. The non-mitochondrial CS from eukaryotes (nmCS) which is part of the glyoxylate cycle[35,36]

branch within the bacterial type II CS, for example. The different CS types (I, II, nmCS) vary in their sequence length and therefore the weight of their monomeric units (Fig. 1b). Between type II and type I CS, our tree features a distinct clade of enzymes from marine Gamma-proteobacteria, Nitrospirota and Cyanobacteria. Enzymes from these groups had not been structurally characterized before but a recent functional analysis had indicated that these form a distinct group of CS[37]. We term this clade type III CS. We have recently described one specific cyanobacterial CS from that clade in detail which assembles into octadecameric complexes (Fig. 1a, *S. elongatus*); details on its evolution and function are described elsewhere[38]. The mitochondrial CS from eukaryotes differ strongly in their sequences forming a very distantly related clade to the bacterial and archaeal CS and were not included in this study.

Guided by our phylogenetic tree, we characterized the quaternary assemblies of CS from extant organisms to represent the diversity of prokaryotic CSs. For 40 CS enzymes across different evolutionary groups we could obtain soluble CS by heterologous expression and found a surprising diversity beyond the known dimeric and hexameric homo-oligomeric complexes (Fig. 1c).

### Hexameric assembly evolved independently in type II and III CS

We first purified enzymes from type III CS and found most of them to form hexamers (Fig. 2a, Supplementary Fig. 2a). The CS from a *Myxococcota* species only assembles into dimers and is found in the first clade that branches off within type III CS (Fig. 2a, b). We found two additional oligomeric state transitions within type III enzymes, both in the group of Cyanobacteria: one is an octadecameric, fractal-like CS, which we described elsewhere[38], and the other one is the loss of the hexameric assembly step in the enzyme of *Cyanobium sp. PCC 7001* (Supplementary Fig. 3a).

We solved a crystal structure for the hexameric type III CS from *M. sulfidovorans* which revealed assembly into dihedral rings of three dimers (Fig. 2c, Supplementary Table 1, Supplementary Fig. 3b). It presented the same type of symmetry as the known hexamers of type II CS and uses broadly the same surface of the dimeric subcomplexes to interact and form complexes. Closer examination of the interfaces revealed that the interfacial residues and molecular interactions differ for hexamers found in type III compared to type II CS (Fig. 2d, Supplementary Fig. 3c). The hexameric complex of *E.coli* (type II CS) is stabilized mostly by van der Waals interactions including the type II-specific JK-loop[22] contacting the opposing dimer (Fig. 2d). This loop is not present in type III CS (Supplementary Fig. 4) and the hexamer from *M. sulfidovorans* instead relies on a cation−π interaction (W150 → R74, K76) and a salt bridge (D151 → R80, Fig. 2d). Exchanging residues of either of these polar interactions disrupts assembly into hexamers resulting in the formation of only dimers (Supplementary Fig. 3d). Many other type III CS apparently employ one or two salt-bridges at the same sites instead of the cation−π interaction found in *M. sulfidovorans* (Supplementary Fig. 3e, Supplementary Fig. 4) indicating a certain flexibility and evolutionary divergence of this interface.

We next asked if type II and III hexamers emerged from a common hexameric ancestor and then diverged substantially or if they represent a case of parallel evolution. To test this, we used ancestral sequence reconstruction (ASR) to resurrect ancient CS that represent the last common ancestor of the two groups (anc$_{2/3}$) as well as subsequent ancestors within type III CS (anc$_{3a}$, anc$_{3b}$, Fig. 2a). The reconstructed sequences were synthesized, heterologously produced and their oligomeric states characterized. Anc$_{2/3}$ does not have the interfacial J/K loop of type II hexamers and is also confidently inferred to lack the interfacial residues of type III hexamers (Supplementary Fig. 5a, b). Consistent with this, it assembled into mostly dimers but we detected a distribution of oligomers within the sample (Fig. 2e). These included predominantly multimers of dimers (tetramers, hexamers, octamers etc.) and also low amounts of oddmers (trimers, pentamers

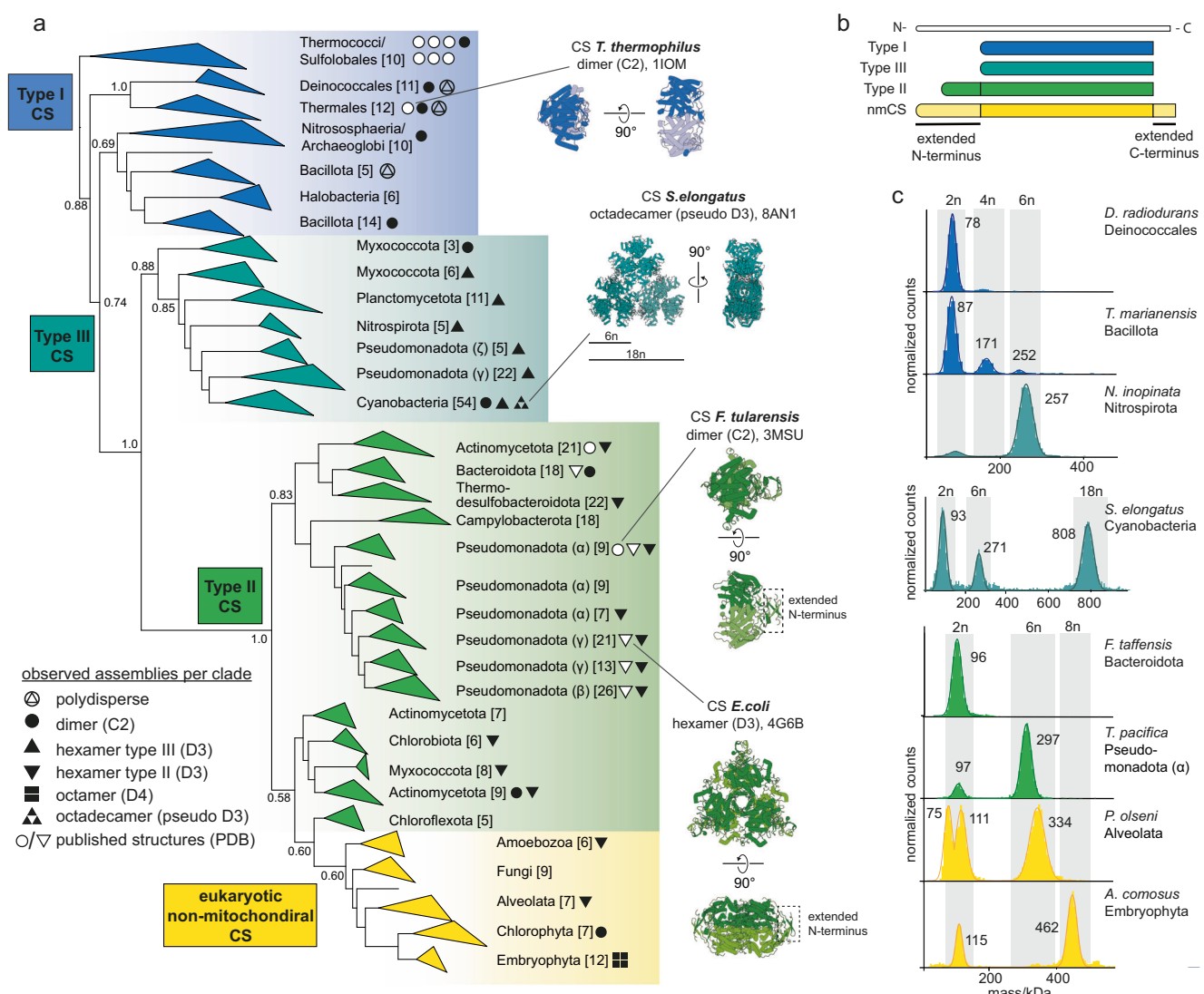

**Fig. 1 | Assembly of citrate synthases across the phylogenetic tree. a** Phylogenetic tree of citrate synthases (CS) in Bacteria, Archaea and eukaryotic non-mitochondrial CS (full phylogeny in Supplementary Fig. 1) and classification into type I, II, III, and nmCS. Brackets indicate the number of sequences within each clade. Branch supports values are shown for important nodes as Felsenstein's bootstrap values. The transfer into Eukaryotes is well supported by phylogenetic and experimental results (see below) but the branching order of major eukaryotic lineages is poorly supported and disagrees with known relationships. Symbols indicate the types of quaternary structure observed within a respective clade for CS. White symbols represent CS structures that have been previously deposited to the Protein Data Bank (PDB) and black symbols correspond to assemblies that were characterized in this study (mass photometry (MP) measurements and PDB accession codes in Supplementary Fig. 2). Representative structures of known CS assemblies are shown. **b** Cartoon representation of the amino acid sequence structure of type I-III and nmCS. **c** MP measurements of purified CS displaying different forms of homo-oligomeric assembly. Source data are provided with this paper.

etc.). This observation is robust to statistical uncertainty in the reconstructed sequence (Supplementary Fig. 6). The larger oligomers are stable and could be isolated via size exclusion chromatography (SEC, Supplementary Fig. 7a). We assigned these oligomeric distributions as polydisperse assemblies. Anc$_{3a}$ revealed a similar behavior as anc$_{2/3}$, again without a preference for an assembly into hexamers (Fig. 2e), which was also recapitulated in an alternative ancestor for this node (Supplementary Fig. 6b). Anc$_{3a}$'s descendant anc$_{3b}$ formed relatively stable hexamers (Fig. 2e). From this we concluded that the type III hexamers evolved in the interval between anc$_{3a}$ and anc$_{3b}$. Our reconstruction is not entirely robust to uncertainty here, with an alternative, lower probability ancestor for this node populating mostly dimers (Supplementary Fig. 6b). To further clarify when hexamers evolved in the type III clade, we purified extant descendants of anc$_{3a}$ and anc$_{3b}$. *Myxococcota* sequences, which descend from anc$_{3a}$ and are sister to all other type III CS, do not have the necessary residues to form the type III interface (Supplementary Fig. 4). The CS we purified from one *Myxococcota* species only forms dimers (Fig. 2b). CS from clades descending from anc$_{3b}$ formed hexamers (Fig. 2a, Supplementary Fig. 2), consistent with the inference from our hexamer-forming maximum aposterori reconstruction of anc$_{3b}$. Our results thus indicate that hexamers evolved independently twice, once within type III CS and once along the lineage to type II CS.

We next wanted to understand how type III hexamers evolved out of a polydisperse ensemble. To evolve to predominantly populate hexamers, anc$_{3b}$ had to stabilize this particular stoichiometry at the expense of the other stoichiometries its predecessor anc$_{3a}$ could populate, including potentially different and asymmetrical hexamers. To understand how this occurred, we introduced individual historical substitutions into anc$_{3a}$ that are in the vicinity of the interface. The location of these amino acids in the protein structure was inferred using a AlphaFold-multimer-v2 prediction of the ancestral CS (Fig. 2f, Supplementary Fig. 7b). Five amino acid substitutions were sufficient to create a CS that lost most of the polydisperse behavior and formed

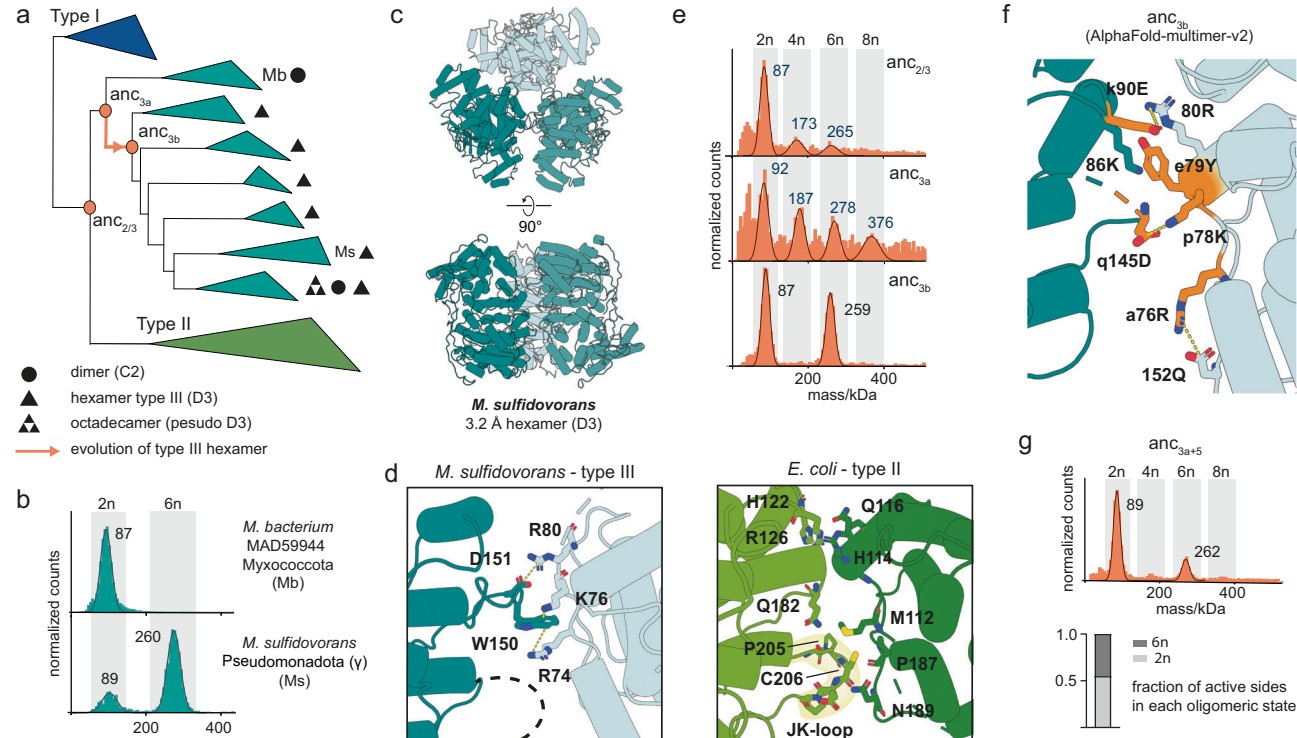

**Fig. 2 | Parallel evolution of hexameric citrate synthases. a** Schematic representation of the citrate synthase (CS) phylogeny displaying the quaternary structures of characterized type III CS. All mass photometry (MP) spectra and species names for the characterized quaternary structures are found in Supplementary Figs. 1, 2. Nodes corresponding to resurrected ancestral CS are indicated (orange circles). **b** MP measurements of two type III CS. **c** X-ray structure of hexameric type III CS from *M. sulfidovorans*. **d** Comparison of the interface area that connects dimers into hexamers in the type III structure and the type II CS from *E. coli*. **e** MP measurements of resurrected ancestral CS along the evolutionary trajectory of hexameric assembly within type III enzymes. **f** Location of historical substitutions (orange residues) within the type III interface in a modelled structure of $anc_{3b}$ using AlphaFold-multimer-v2. **g** MP measurement a variant of $anc_{3a}$ with a set of 5 historical substitutions shown in (**f**) that are sufficient to trigger formation of hexamers and loss of polydisperse behavior ($anc_{3a+5}$). Bar graph displays the fraction of all active sites in dimers vs hexamers for $anc_{3a+5}$. Source data are provided with this paper.

moderate amounts of hexamers ($anc_{3a+5}$ Fig. 2g, Supplementary Fig. 7c). Three of the substitutions result in the creation of two salt bridges between dimers (q145D → p78K, k90E → 80 R, Fig. 2f; small and capital letters indicate the ancestral and derived amino acid, respectively). The e79Y substitution introduces an aromatic side chain which is highly conserved in type III enzymes and probably stacks against the backbone of a conserved lysine (K86) (Fig. 2f; Supplementary Fig. 4). The last substitution introduced an intramolecular interaction within (a76R→152Q) which was apparently necessary to stabilize hexamers. Interestingly, in the CS from *M. sulfidovorans* this arginine is recruited into the cation–π interaction that stabilizes the hexameric interaction (R74, Fig. 2d).

Together, these results show that hexamers evolved through introduction of salt bridges and other polar interactions between dimers. Salt bridges alone usually do not confer strong binding energy because of the desolvation costs of their charged moieties[39]. But they are known to introduce specificity into interactions[40,41]. A plausible mechanism is thus that these 5 substitutions introduce specific contacts that favour symmetrical hexamers, whilst relying on prior affinity between dimers in the polydisperse ensemble.

### Interface turnover in type II citrate synthases

We next investigated the history of hexamers in type II CS (Fig. 3a). All type II CS have the JK-loop that is part of the interface that holds together dimers into hexamers in published type II CS structures (Supplementary Fig. 8a). Most of the enzymes we characterized from different clades within type II CS assemble into hexamers (Fig. 3a, b, Supplementary Fig. 2a). Additionally, we identified two extant type II

CS that only formed dimers (Fig. 1a *F. taffensis*, Fig. 3b *C. woesi*). This oligomeric state had been observed for the type II CS of *F. tularensis* and *M. tuberculosis* before (Fig. 3a, white circles). These dimers represent four independent reversions from hexamers, which occurred in distant parts of the type II phylogeny (Fig. 3a).

Using ASR, we could not identify the exact time point when the hexameric assembly first emerged in type II CS. We found the last common ancestor of all type II CS ($anc_2$) to be polydisperse, even though it contains the characteristic J/K loop of type II hexamers as well as its binding-site on the opposing dimer (Fig. 3c). We then characterized $anc_2$'s two daughter nodes in the type II phylogeny. One ($anc_{2b}$) assembled into weak hexamers but not the other ($anc_{2a}$, Fig. 3c). More recent ancestral proteins descending from $anc_{2a}$ were also ambiguous regarding their assembly into hexameric complexes, as were alternative reconstructions of these ancestors (Supplementary Fig. 8b, c, Supplementary Fig. 6b). Taken at face value, this would indicate that the specific hexamers emerged multiple times within type II CS using the same residues. Unlike our earlier inference about the LCA of type II and III CS, this is very unparsimonious with respect to the stoichiometries of extant CS: we found hexamers all across type II CS, including in each descendant lineage of all deep type II ancestors with all using the JK-loop as part of their interface (Supplementary Fig. 8d). The network of interactions in the interface of type II hexamers diverges relatively strongly in different phyla (Supplementary Fig. 9). This makes accurate reconstruction of this interface potentially challenging: mis-matched pairs of interacting residues could potentially destabilize the interface and be the reason why we only found the hexameric assembly in $anc_{2b}$. We therefore think hexamerization likely

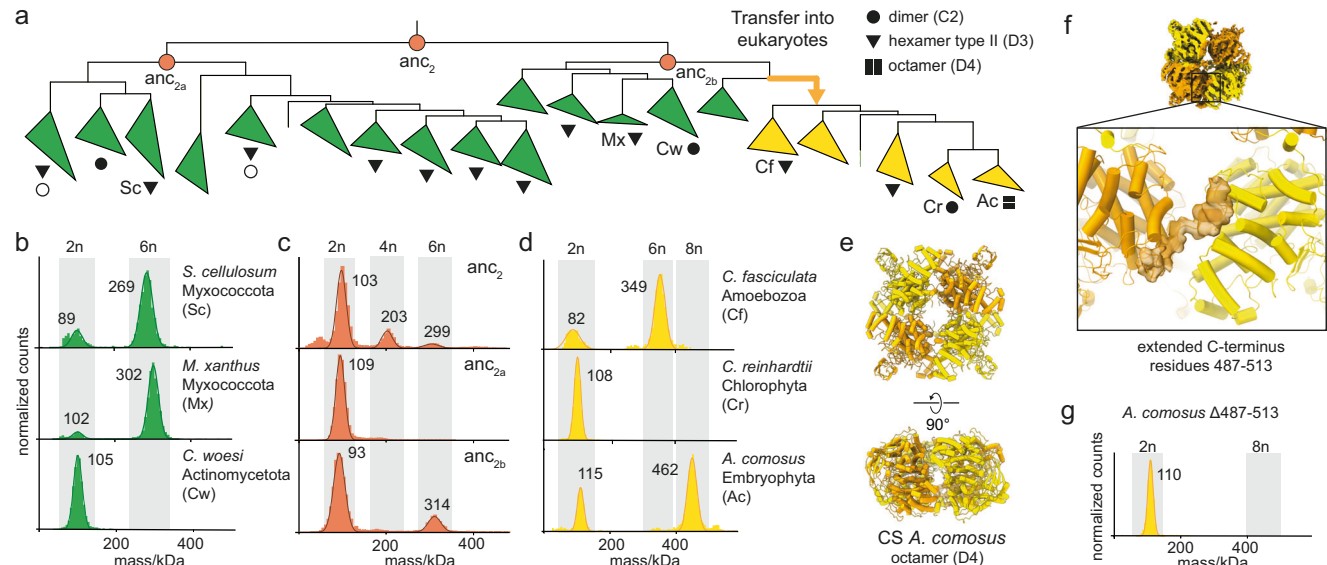

**Fig. 3 | Widespread hexameric assembly in type II citrate synthases and evolution of novel octamers. a** Schematic representation of part of the citrate synthase (CS) phylogeny displaying the quaternary structures of characterized type II CS and non-mitochondrial CS (nmCS). All mass photometry (MP) spectra and species names for the characterized quaternary structures are found in Supplementary Figs. 1, 2. Nodes corresponding to resurrected ancestral CS are indicated (orange circles). **b**–**d** MP measurements of extant type II CS (**b**), ancestral type II CS (**c**) and nmCS (**d**). **e** Cryo-EM structure of octameric nmCS from *A. comosus*. **f** Focus on the extended C-terminus of nmCS within the cryo-EM density of *A. comosus* which folds over from one dimeric subcomplex to an adjacent one. **g** MP measurement of a variant of *A. comosus* CS in which the extended C-terminus was deleted (Δ487-513). Source data are provided with this paper.

emerged early in type II CS evolution and was subsequently lost at least four times resulting in dimeric enzymes in several extant species.

Eukaryotic nmCS were phylogenetically inferred to originate from type II CS. Purification of two nmCS enzymes revealed the typical hexameric assembly and corroborated this inference (*C. fasciculata* Fig. 3d, *P. olsenii* Fig. 1c). All nmCS also have the JK-loop that forms the interface between dimers in type II CS (Supplementary Fig. 9) but not all homologs retained hexameric assembly: the enzyme from the unicellular algae *Chlamydomonas reinhardtii* reverted to forming only dimers (Fig. 3d). We discovered another structural type of CS within nmCS: the enzyme from the land plant *Ananas comosus* mainly assembled into octamers (Fig. 3d). We solved a medium-resolution cryo-EM structure for the complex and identified the octamers to be dihedral rings of four dimers (Fig. 3e, Supplementary Table 2, Supplementary Fig. 10). The interactions between dimers that form the octamer are different from type II or III hexamers. The interaction is made by the C-terminus of the protein, which is considerably elongated in nmCS compared to type I-III CS (Fig. 1b) and folds from one dimer over to an adjacent one (Fig. 3f). No other evident interactions between dimers were observed as their main chains appeared to be too far apart. To verify this observation in light of the modest resolution of the structure we deleted the C-terminal extension. This truncated CS protein (*A. comosus* Δ487-513, Fig. 3g) could only assemble into dimers. This implies that the contacts that hold other type II CS into hexamers are completely lost in this protein and replaced by a new interaction through the C-terminus. To understand how this occurred, we inferred an AlphaFold-Multimer prediction for the hexameric nmCS of *C. fasciculata*. The predicted structure indicated that its elongated C-terminus could already connect to the adjacent chain while still having the type II interface contacts to form hexamers (Supplementary Fig. 8d, e). This suggested that a switch in interfaces could have emerged via an evolutionary intermediate that had two interfaces: one via the C-terminus and one via the JK-loop. This redundancy would have then allowed the old interface to be lost somewhere in the Embryophytes, and presents a possible trajectory towards the to the evolution of octamers held together exclusively by their C-termini.

## Assembly into a polydisperse distribution is an ancient trait of CS

We observed CS to assemble into a polydisperse distribution of oligomeric states in the ancestors of both type II (anc$_2$) and type III CS (anc$_3$) as well as their common ancestor (anc$_{2/3}$). To understand the evolutionary origin of this behavior and if it could be a transition state from dimeric to hexameric quaternary structure, we resurrected and characterized the ancestors of different clades of type I CS (Fig. 4a). All of these ancestral CS were also polydisperse (anc$_{1a-c}$, Fig. 4b) indicating that this type of assembly is probably very ancient since these represent precursors of phyla that are thought to have diverged early in microbial evolution[30,31]. To confirm that this is not an artefact arising from incorrectly reconstructed sequences, we resurrected alternative, less likely sequences for these ancestors. The resulting proteins differed in 55–69 amino acid residues from the most-likely ancestral sequences but still retained the polydisperse phenotype although weaker to some degree (altall anc$_{1a-1c}$, Supplementary Fig. 6). In addition, we asked if this assembly form is also found in any early branching type I CS. We indeed found multiple extant type I CS that branch off early in their respective clades closely to these ancestors and also populate polydisperse ensembles that are stable and could be isolated via SEC (Fig. 4c, Supplementary Fig. 11a).

We next sought to better understand the organization of these oligomeric ensembles to test if they represent structural precursors out of which type II and type III hexamers emerged. Because the polydisperse nature of these proteins makes high-resolution structure determination challenging, we separated them by size via SEC to enrich for large stoichiometries. We were able to enrich anc$_{1b}$ for hexamers but still found the sample to be highly heterogeneous when analyzing it via cryo-EM. This hindered obtaining high-resolution structural insights into this assembly but we could retrieve 2D class averages that appear to correspond to hexameric assemblies (Supplementary Fig. 11b–d). From these it appears that the dimeric subcomplexes connect to each other without forming closed circular assemblies, resulting in the formation of non-symmetric higher-order oligomers.

Our data thus imply that the most ancient forms of CS were capable of forming polydisperse ensembles of potentially asymmetric

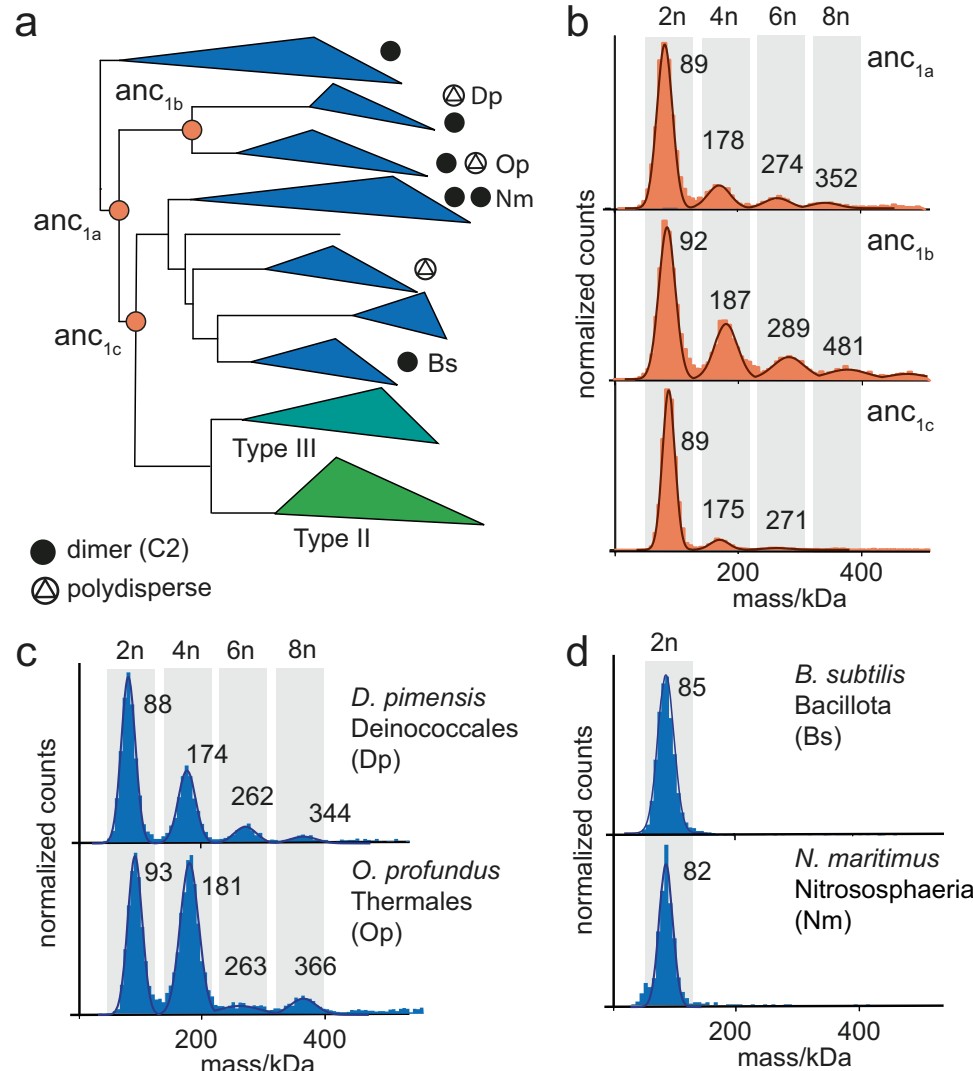

**Fig. 4 | Polydisperse assemblies early in the evolution of citrate synthase.**
**a** Schematic representation of the citrate synthase (CS) phylogeny displaying the quaternary structures of characterized type I CS. All mass photometry (MP) spectra and species names for the characterized quaternary structures are found in Supplementary Fig. 1-2. Nodes corresponding to resurrected ancestral CS are indicated (orange circles). **b**–**d** MP measurements of ancestral CS (**b**), polydisperse (**c**) and dimeric (**d**) type I CS. Source data are provided with this paper.

larger order complexes. Most type I CS then lost this ability and became monodisperse dimers (Fig. 4a, d). Similarly, type II and type III CS also became monodisperse, but settled on two different types of hexamers instead of dimers. We presently have no explanation for this repeated evolution of more monodisperse assemblies, but it may indicate either mutational or selective pressure (or a combination of both) towards monodispersity.

**Oligomeric state transitions lead to minor changes in catalytic parameters**

We have shown that there are many different quaternary structures that CS can assemble into, but all of these proteins catalyze the same enzymatic reaction. We therefore examined if changes in oligomeric state are connected to differences in catalytic parameters, which selection could in principle act on. First, we characterized the catalytic activity of the different oligomeric forms found within polydisperse CS. Here all complexes are built from the same sequence. We separated the polydisperse complexes of the CS from *D. pimensis* via SEC and isolated fractions with different oligomeric compositions (Fig. 5a, Supplementary Fig. 11a). The fractions varied from almost exclusively dimeric to a majority of complexes with ≥ 8 subunits and their assemblies did not re-

equilibrate for several hours. We then measured Michaelis-Menten kinetics of the individual fractions. This revealed that all complexes are catalytically competent enzymes. There is a reduction in the turnover number $k_{cat}$ for larger oligomeric complexes compared to the dimers (Fig. 5b), which may derive from steric hindrance of important catalytic motions. The $K_m$-values in contrast are very similar for all measured fractions (Supplementary Table 3). The results illustrate that different oligomeric species can have somewhat different catalytic properties, even if their sequences are identical. The catalytic parameters of the full equilibrium of the protein when not enriched for larger stoichiometries (Dp) was, however, very similar to the one of pure dimers. Considering the catalytic properties, an evolutionary transition from polydisperse to dimeric state or vice versa therefore might not be visible to selection in this case, because the fraction of active sites in the larger oligomers is small enough not to have a major effect.

In another case, different oligomeric states appear not to differ in their catalytic activity: We discovered that we could shift the oligomeric state of the CS from *A. comosus* from octamers to roughly similar amounts hexamers and octamers with high quantities of acetyl-CoA (but not oxaloacetate) (Fig. 5c). In principle, this could mean that hexamers are the catalytically competent form. Further experiments

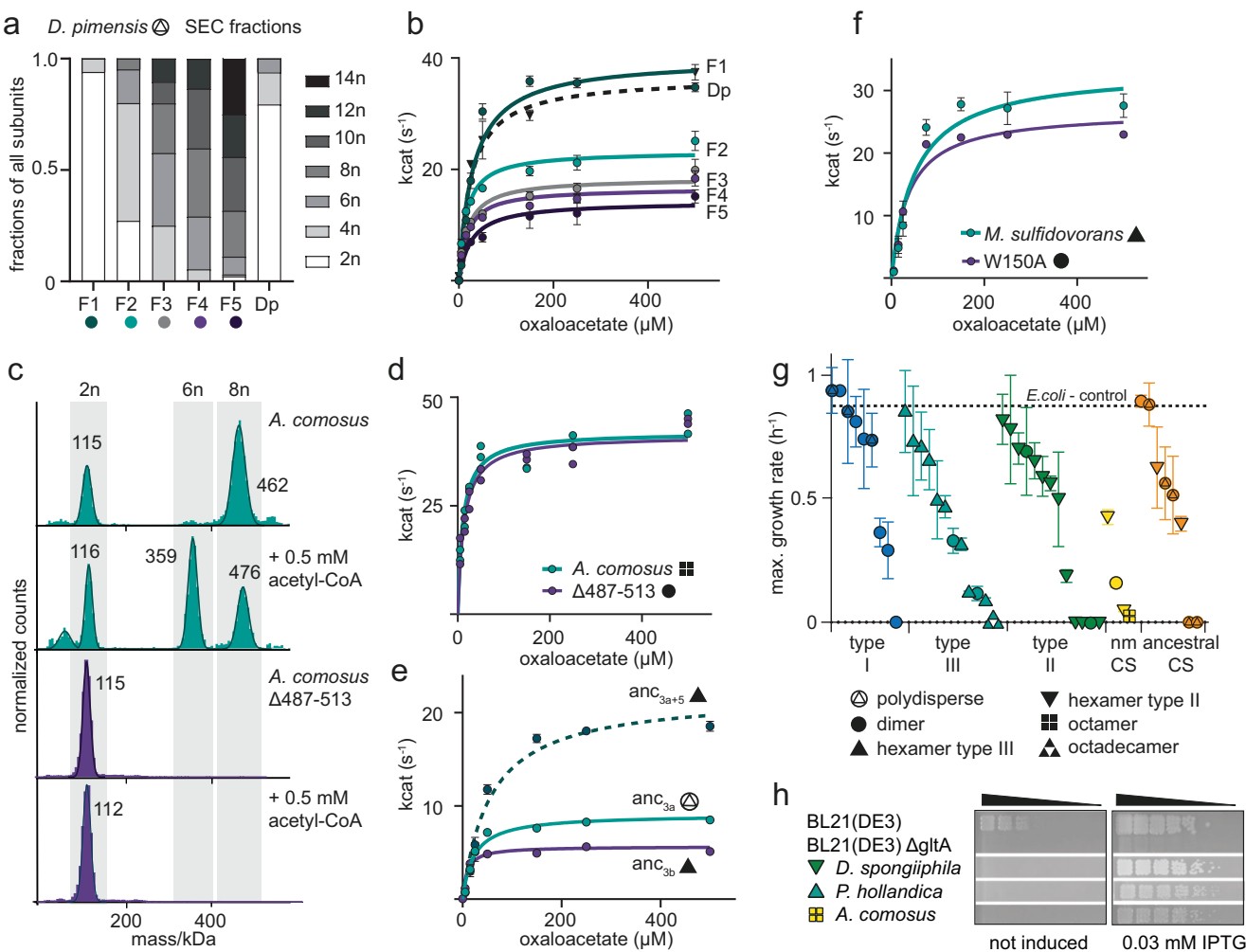

**Fig. 5 | Effects on catalytic function by changes in quaternary structure.**
**a** Fraction of citrate synthase (CS) subunits within the different oligomeric states for each size exclusion chromatography (SEC) fraction determined by mass photometry (MP) for the CS from *D. pimensis* (Dp) (F1-F5, see also Supplementary Fig. 11a). **b** Michaelis-Menten kinetics of the different fractions shown in (**a**) and the unseparated sample of *D. pimensis* CS (Dp). Data are presented as mean values +/− SD, *n* = 3 technical replicates. **c** MP measurements of CS from *A. comosus* and its truncation variant Δ487-513 in the absence and presence of acetyl-CoA. **d** Michaelis-Menten kinetics of CS from *A. comosus* and its Δ487-513 variant, *n* = 2 technical replicates. **e** Michaelis-Menten kinetics of ancestral CS bracketing the emergence of hexamers within type III enzymes (anc$_{3a}$, anc$_{3b}$) and the minimal substitution construct to yield hexameric complexes (anc$_{3a+5}$). Data are presented as mean values +/− SD, *n* = 3 technical replicates. **f** Michaelis-Menten kinetics of extant CS from *M. sulfidovorans* and a variant that disrupts the interface between dimers (W150A). Data are presented as mean values +/− SD, *n* = 3 technical

replicates. **g** Maximum growth rates on M9 media of an *E. coli* strain lacking the native CS gene (BL21(DE3) ΔgltA) complemented with a plasmid encoding for CS genes from all characterized extant and ancestral enzymes. Data are presented as mean values +/− SD, *n* = 3 biological replicates. The positive control is the CS knockout strain complemented with the native CS from *E. coli* which displayed the same growth behaviour as the wild-type BL21(DE3) strain. **h** Spot assays on solid M9 media using either leaky expression or induction by isopropyl β- d-1-thiogalactopyranoside (IPTG). Representative strains are depicted that did not complement in (**g**), but showed complementation upon increased CS production. Cultures were spotted in a five-step serial dilution using a ratio of 1:5 for each step and incubated on M9-solid media supplemented with glucose and different IPTG concentrations. One representative plate is shown for each experiment, out of a total of three replicates for each plate. Full data for all strains are shown in Supplementary Fig. 14. Source data are provided with this paper.

contradict this: We compared the Michaelis-Menten-kinetics of the *A. comosus* Δ487-513 variant, which can only form dimers, to that of the wild-type enzyme and found them to be highly similar (Fig. 5d, kinetic parameters for oxaloacetate were measured at saturating concentrations of 500 μM acetyl-CoA; Supplementary Fig. 12a). It thus appears that this CS has evolved substrate-dependent oligomeric state changes that are not required for catalysis, though they could plausibly be important for other physiological regulatory processes like allosteric regulation by small molecules, interactions with other enzymes of the TCA cycle, or representing a storage form, which we did not investigate here.

We next asked if the evolution of a new interface via changes in the sequence had effects on CS kinetic parameters. We decided to

investigate the transition within type III enzymes from dimers to hexamers because here we have two ancestral enzymes bracketing this transition (anc$_{3a}$, anc$_{3b}$). We found the hexameric anc$_{3b}$ to have a slightly lower k$_{cat}$ and a reduced K$_m$-value compared to the polydisperse anc$_{3a}$ (Fig. 5e, Supplementary Fig. 12b, Supplementary Table 4). Both effects are very small compared to the variation in enzymatic parameters for bacterial CS[42]. We next asked if this change is causally linked to the substitutions that caused hexamers to evolve. We characterized the variant of anc$_{3a}$ in which we transplanted five historical substitutions that were sufficient to trigger hexamer formation (anc$_{3a+5subs}$). In this variant roughly 45% of active sites reside in hexamers at assay concentrations (Fig. 2g). The k$_{cat}$ of this variant was more than twofold higher than both of the ancestors and the K$_m$-value

was also elevated (Fig. 5e, Supplementary Fig. 12b, Supplementary Table 4). The mild decrease in activity between $anc_{3a}$ and $anc_{3b}$ is therefore likely not linked to the gain of an interface but to some of the other 24 substitutions that occurred in that interval. This demonstrates that a transition in quaternary structure can be accompanied immediately by a change in catalytic activity (in this case an increase in $k_{cat}$). But in this instance, this effect was almost immediately reversed by additional substitutions outside of the interface. In addition, all three enzymes ($anc_{3a}$, $anc_{3+5}$, $anc_{3b}$) have very similar rates of catalysis at substrate concentrations below < 25 µM substrate. If intracellular substrate concentrations are low, which for oxaloacetate is usually expected due to its instability[43], the changes in maximum turnover number ($k_{cat}$) would likely not be visible in selection. In addition, we quantified the effect on thermal stability of this particular shift towards hexamers. We found that $anc_{3b}$ and $anc_{3+5}$ had slightly lower melting points than $anc_{3a}$ (77.6 °C and 76.8 °C, compared to 81.3 °C, Supplementary Fig. 12c). This implies that adding interfaces at least initially does not necessarily raise stability.

Lastly, we investigated if the loss of an assembly step causes a change in catalytic activity. This may be the case if interfaces have accumulated substitutions that are harmless when the complex can assemble, but highly deleterious when it is prevented from forming. Such 'entrenchment' for example occurs if interfaces become so hydrophobic that they need to be shielded from solvent to prevent aggregation[44–46]. To test for this, we abolished interfaces of an extant oligomeric complex by mutation. We measured kinetic parameters of the type III hexamer from *M. sulfidovorans* and its dimeric interface mutant (W150A) and found them to be very similar (Fig. 5f, Supplementary Fig. 12d). We also measured the thermal stability of this variant and compared it to the stability of the wild-type protein, which had a melting point of 67.0 °C. The W150A mutant unfolded faster, in a two-step process with a melting point of 49.3 °C for the first step and the second step corresponding more closely to the unfolding curve of the wild-type (Supplementary Fig. 12e). This implies that the this CS has accumulated some substitutions that entrench this interface, but this effect only manifests at temperatures far above the optimal growth temperature of the microbe this CS is found in (around 22 °C[47]). At physiological growth temperatures, which we also used in our assays this does not lead to a measurable loss of activity. Taken together, these results imply that at least in this case a new interface had a minor effect on catalysis and that its loss has effects on stability that are tolerable at physiological temperatures. Similar principles may explain the relatively frequent losses of higher-order assembly we observed on our phylogeny.

**Many different CS can substitute for an allosterically regulated hexamer CS in *E. coli***

Last we investigated the ability of different types and oligomeric forms of CS to complement its native function in vivo. We chose *E. coli* as host which natively holds a hexameric type II CS, that can be allosterically inhibited by NADH[25]. To do this we constructed a scarless knock-out (KO) strain which lacked the native CS gene (*E. coli* BL21(DE3) Δ*gltA*). This strain can grow on complex media but is not viable if only glucose or acetate is available as sole carbon source because of the dependency on CS activity. We transformed this strain with plasmids encoding CS from different extant organisms or ancestral CS and recorded growth curves in minimal media (M9) supplemented with glucose (Supplementary Fig. 13). We found that CS function in *E. coli* can be complemented by most of our enzymes to some degree and that there is no obvious correlation to type or quaternary structure (Fig. 5g). All polydisperse and many dimeric type I CS produced growth rates comparable to complementation with the native enzyme (Fig. 5g). For type II and III CS, some complement well but others do not – but there is no indication that hexameric CS in general or specifically type II hexamers perform better in *E. coli* than other types of CS. All characterized nmCS complement poorly but this is likely

connected to low levels of protein production. In general, we found that the protein production of the different enzymes varied very strongly and that the CS that do not complement in growth curve experiments are often not efficiently produced (Supplementary Fig. 14a). If we increase the expression by addition of an inducer (IPTG) eventually all tested CS can promote growth of the KO-strain at least on solid media (Fig. 5h, Supplementary Fig. 14b). This demonstrates that all tested CS retain the ability to rescue CS function in *E. coli* independently of primary or quaternary structure at least under strong selection in a laboratory setting. Self-assembly type is thus not a strong determinant of the adaptive value of any particular CS to *E. coli* even though its native CS is an allosterically regulatable hexamer.

## Discussion

Here we have shown that even within the evolutionary history of only one protein family there is remarkable diversity in the assembly into homo-oligomeric complexes. This echoes previous findings from Form II Rubiscos, which also transitions between different oligomeric states[17]. In particular, we have discovered several types of assemblies that are unlikely to be well represented in the protein data bank[48]: enzymes that can adopt multiple interconverting and likely asymmetric assemblies; complexes that form remarkable fractal-like structures[38], and one oligomer held together by very flexible terminal interactions. In addition, all these assembly types (with perhaps the exception of the octamer) have been lost at least once and in several cases multiple times. Our discovery rate of oligomeric state transitions was remarkably high for such a conserved enzyme, and it is entirely possible that more surprises lurk in the CS phylogeny.

Some generalizable principles emerge: First, all assemblies we have structures of are either dimers or rings of dimers, in which the active site is located on the outward-facing side of the dimer. This almost certainly reflects the constraint that any oligomeric assembly should not block access to the active site. A similar argument may also explain another bias: CS appears to have only evolved ring-like assemblies of at least three dimers. We never observed dihedral dimers of dimers. We can only speculate that such arrangements, which are very common in other families[49,50], never evolved in CS because they in some way sterically hinder CS catalytic motion[51,52], though sheer coincidence remains a plausible explanation. A third conspicuous pattern is the frequent loss of higher-order assembly. This implies that in many cases the interfaces that hold such assemblies together are not entrenched. Structurally, the reason may be that these interfaces are in general quite small and relatively hydrophilic, especially when compared to larger and very hydrophobic dimer interface in CS[46].

Perhaps our most surprising discovery is that multiple CS on our tree can populate several stoichiometries at once and that this trait has been lost multiple times in favor of more monodisperse assemblies. We at present have no good explanation for why this is the case. One hypothesis would be that sequences of monodisperse quaternary structures are evolutionary more readily available than the ones of the polydisperse enzymes. Polydisperse assembly is not known to be common in other protein families[53], but that may result from a discovery bias: Most traditional techniques for measuring quaternary assemblies struggle to resolve such ensembles. In either case, these assemblies seem to influence kinetics only mildly and may thus represent a form of harm- and useless complexity.

Is all this stoichiometric variation functionally meaningless? This is difficult to answer without detailed experiments in the organisms that host these enzymes. We did this recently for the fractal-like CS from the cyanobacterium *S.elongatus* and could not detect any functional advantage of this bizarre assembly over a simpler enzyme in vivo[38]. We thus think it at least plausible that some of these stoichiometries have no particular function. On the other hand, even initially useless evolutionary inventions can be the seed of future adaptive functions that are built on top of them. One example is likely

the allosteric regulation via NADH in type II CS, which is found only in a subset of type II hexamers. Overall, we suspect that many proteins and specifically enzymes can wander quite freely through the space of possible multimeric states. This observation agrees with recent computational inferences[54] and raises the question of what our null hypothesis should be when we discover a new type of quaternary assembly. Based on the work presented here, we advocate to assume it is useless until it is proven not to be.

## Methods

### Molecular cloning

The genes encoding the CS from *Bacillus subtilis*, *Escherichia coli* and *Myxococcus xanthus* were amplified from genomic DNA by PCR (Q5® High-Fidelity 2X Master Mix, New England Biolabs, USA-MA) and introduced into the pLIC expression vector[55] by Gibson cloning (Gibson Assembly Master Mix, New England Biolabs, USA-MA). All other extant and ancestral CS sequences were obtained as gene fragments from Twist Bioscience (USA-CA) or Integrated DNA Technologies (USA-IA) and introduced in the same expression vector by Gibson cloning. All CS sequences were tagged with a C-terminal polyhistidine-tag for purification (tag-sequence: LE-HHHHHH-Stop). For single-site mutants and deletions of the CS-sequences the KLD Enzyme Mix (New England Biolabs, US-MA) was used. Mutagenesis primers were designed with NEBasechanger (nebasechanger.neb.com) and used to PCR-amplify the vector encoding for the gene that was to be changed. Resulting PCR products were added to the KLD enzyme mix and subsequently transformed. All cloned genes were verified by Sanger-sequencing (Microsynth, Germany).

The DNA sequences of all purified proteins and the NCBI identifier of all extant sequences can be found in the Supplementary Data 1.

### Protein purification

For heterologous overexpression the vectors with the gene of interest were transformed into chemically competent *E. coli* BL21 (DE3) cells. Transformed colonies were used to inoculate expression cultures (500 mL) made from LB-medium supplemented with 12.5 g/L lactose (Fisher chemical, USA-MA). The cultures were incubated overnight at 30 °C and 200 rpm. Cells were harvested by centrifugation (4500 x g, 15 min, 4 °C), resuspended in Buffer A (20 mM Tris, 300 mM NaCl, 20 mM imidazole, pH 8) and freshly supplemented with DNAse I (3 Units/µL, Applichem GmbH, Germany). Cells were disrupted using a Microfluidizer® (Microfluidics International Corporation, USA-MA) in three cycles at 15.000 psi and centrifuged to spin down cell debris and aggregates (30,000 x g, 30 min, 4 °C). The clarified lysate was loaded with a peristaltic pump (Hei-FLOW 06, Heidolph, Germany) on pre-packed nickel-NTA columns (5 mL Nuvia IMAC Ni-Charged, Biorad, USA-CA) that were equilibrated before with buffer A. The loaded column was first washed with buffer A for seven column volumes and then with 10% (v/v) buffer B (20 mM Tris, 300 mM NaCl, 500 mM imidazole, pH 8) in buffer A for seven column volumes. The bound protein was eluted with buffer B and either buffer exchanged with PD-10 desalting columns (Cytiva, USA-MA) into PBS or 20 mM Tris, 200 mM NaCl, pH 7.5 or further purified by size-exclusion chromatography (SEC). For SEC the protein was injected on an ENrich SEC 650 column (Biorad, USA-CA) with PBS as running buffer using a NGC Chromatography System (Biorad, USA-CA). To separate the different oligomers of polydisperse assemblies fractions of 250 µL were collected into a 96-well plate and analyzed via MP afterwards. Purity of the proteins was analyzed by SDS-PAGE. After either buffer exchange or SEC, the purified proteins were flash-frozen with liquid nitrogen and stored at −20 °C before further use.

### Phylogenetic analysis and ancestral sequence reconstruction

Amino acid sequences of 418 CS genes from bacteria, archaea and eukaryotes were collected from the NCBI Reference Sequence Database and aligned via MUSCLE v3.8.31[56]. The maximum likelihood (ML) phylogeny was inferred from the multiple sequence alignment (MSA) using raxML v8.2.10[57]. The LG substitution matrix[58] was used as determined by automatic best-fit model selection as well as fixed base frequencies and a gamma-model of rate heterogeneity. The robustness of the ML tree topology was assessed by inferring 100 non-parametric bootstrap trees with raxML, from which Felsenstein's and transfer bootstrap values were derived using BOOSTER (booster.pasteur.fr). Using PhyML 3.0[59], we also inferred approximate likelihood-ratio test (aLRT)[60] for branches to statistically evaluate the branch support in the phylogeny.

Based on the CS tree and the MSA, ancestral sequences were inferred using the codeML package within PAML v4.9[61]. To adjust for gaps and the different length of the N-termini of the CS sequences, their ancestral state was determined using parsimony inference in PAUP 4.0a based on a binary version of the MSA (1=amino acid, 0=gap, no residue). The state assignment for each node in the tree (amino acid or gap) was then applied to the inferred ancestral sequences. Alignment figures were created using ESPrit3.0[62]. To test the robustness of the ancestral proteins we additionally inferred alternative ancestors (altall). These sequences contain the second most likely reconstructed amino acid for each state for which it has a PP > 0.2. All other states present the ML state yielding a "worst case" scenario ancestor. For altall $anc_{3a}$ and altall $anc_{3b}$ the threshold was set to PP > 0.3 because the more stringent threshold resulted in instable, monomeric proteins.

### Mass photometry

Measurements were performed on a OneMP or a TwoMP mass photometer (Refeyn Ltd, UK). Reusable silicone gaskets (CultureWellTM, CW-50R-1.0, 50-3 mm diameter x 1 mm depth) were set up on a cleaned microscopic cover slip (1.5 H, 24 × 60 mm, Carl Roth, Germany) and mounted on the stage of the mass photometer using immersion oil (IMMOIL-F30CC, Olympus, Japan). The gasket was filled with 19 µL buffer (PBS or 20 mM Tris, 200 mM NaCl pH 7.5) to focus the instrument. Then, 1 µL of prediluted protein solution (1 µM) was added to the buffer droplet and mixed thoroughly. Final concentration of the proteins during measurement was between 12.5 and 50 nM. Data was acquired for 60 s at 100 frames per second using AcquireMP (Refeyn Ltd, v1.2.1). The resulting movies were processed and analyzed using DiscoverMP (Refeyn Ltd, v2022 R1). The instrument was calibrated at least once during each measuring session using a homemade calibration standard of a protein mixture with known sizes (86–430 kDa).

For measurements with substrates/effector molecules the prediluted protein sample (2 µM) was incubated for 10 min with the respective effector concentration. The same substrate concentration was also included in the buffer in the gasket that was used for focusing.

### Kinetic enzyme assays

For the CS kinetic assays the colorimetric quantification of thiol-groups was used based on 5,5′-dithiobis-(2-nitrobenzoic acid) (DTNB)[63,64]. The photospectrometric reactions were carried out in 50 mM Tris pH 7.5, 10 mM KCl, 0.1 mg/mL DTNB and 25 nM protein concentration at 25 °C. To measure $K_m$ values, one substrate was saturated and added to the reaction mix (1 mM oxaloacetate or 0.5 mM acetyl-CoA respectively). The other substrate was varied in concentration and added last to start the reaction (25, 50, 75, 100, 150, 250 and 500 µM respectively). For enzymes with low $K_m$ values the reaction velocity was additionally measured at 5 and 15 µM substrate. Reaction progress was followed by measuring the appearance of 2-nitro-5-thiobenzoate at 412 nm (Extinction coefficient 14.150 $M^{-1}$ $cm^{-1}$) in a plate reader (Infinite M Nano + , Tecan, Switzerland). Data analysis and determination of enzyme kinetic parameters was done with GraphPad Prism (Version 8.4.3).

### Circular Dichroism spectroscopy

For CD spectroscopy, proteins were prepared at 1 mg/mL in buffer containing 20 mM Tris, pH 7.5, 100 mM NaF, and 8% glycerol. All data

was collected on a Jasco J-810 Spectropolarimeter using a Hellma Macro-cuvette 110-QS (1 mm layer). For each sample a spectral run was performed over the temperature range from 20 °C to at 95 °C in 5 °C increments, with measurements at each temperature following a 3 min equilibration period. At each temperature three repeat spectra were collected over the wavelength range from 280 to 190 nm in 1 nm steps. Scans of each sample were scaled and single wavelength melt curves were obtained by plotting measured values of the 223 nm peaks of each spectrum against temperature. $T_m$ values (the temperature point where 50% of protein unfolding occurs) were determined directly from the plot. For the two-step conversion the melting curve was separated at 65 °C. The two separate melting curves were scaled and plotted so that the $T_m$ value for both melting events could be estimated.

### Crystallography and structure determination

A solution of 20 mg/mL CS from *M. sulfidovorans* was incubated with 5 mM oxaloacetic acid. Crystallization was then performed by the hanging-drop method (Crystalgen 24-well plate, pregreased - microscpe cover slips as lid) in 1 μL drops consisting of equal parts of protein and precipitation solutions. Diamond shaped crystals formed after 20 weeks at 20 °C. The crystallization condition consisted of 1 M LiCl, 0.1 M citric acid and 10% PEG 6000 at a pH of 5.0. For data collection, a cryo solution consisting of 70% mother liquor and 30% glycerol was added and the crystal was flash-frozen in liquid nitrogen. Data were collected at 100 K at Deutsches Elektronen-Synchrotron (Hamburg, Germany). MxCube2 was used for data collection[65], yielding a dataset with 98.85% completeness. Data were processed with XDS (version 06/2023) and scaled with XSCALE[66]. The structure was initially determined by molecular replacement with PHASER[67], utilizing a monomer generated in silico by AlphaFold2[68]. The structure was then iteratively built in WinCoot (version 0.9.6)[69] and refined with PHENIX (version 1.19)[70].

### Cryo-electron microscopy

For cryo-EM sample preparation of CS from *A. comosus*, 4.5 μl of the purified protein at 1 mg/mL were applied to glow discharged Quantifoil R2/1 Cu 200 mesh grids, blotted for 3.5 s with force 4 in a Vitrobot Mark IV (Thermo Fisher) at 100% humidity and 4 °C, and plunge frozen in liquid ethane, cooled by liquid nitrogen. Cryo-EM data was acquired with a FEI Titan Krios transmission electron microscope using SerialEM software[71]. Movie frames were recorded at a nominal magnification of 29,000X using a K3 direct electron detector (Gatan) operating in electron counting mode. The total electron dose of ~55 electrons per Å² was distributed over 30 frames at a calibrated physical pixel size of 1.09 Å. Micrographs were recorded in a defocus range of −0.5 to −3.0 μm.

For each sample of polydisperse anc$_{1b}$ at 0.25 mg/mL 4 μL of protein suspension was applied onto glow discharged Quantifoil grids (R 1.2/1.3 Cu 200 mesh). All samples were plunge frozen in a propane/ethane (63%/37%) mixture using a ThermoFisher Vitrobot Mark IV at 4 °C, 100% humidity and a blotting time of 5 s. For the first data set blot force was set to a value of 4 whereas for data set two and three a value of 6 was used. Cryo-EM data of the polydisperse anc$_{1b}$ sample was collected on a CRYO ARM 200 (JEOL) transmission electron microscope (TEM) operated at 200 kEV and equipped with a K2 direct detector (Gatan). Three datasets were recorded at a magnification of 60,000× corresponding to a pixel size of 0.85 Å/pixel and with a total dose of 50 e/Å² for dataset 1 (2646 movies, 50 frames), 40 e/Å² for both dataset 2 (5258 movies, 40 frames), and dataset 3 (1569 movies, 25 frames). SerialEM[71] was used for automated data acquisition and micrographs were pre-processed with CryoSPARC Live[72].

### Image processing, classification and refinement

For the CS from *A. comosus* all processing steps were carried out in cryoSPARC v3.1.0[72] (Supplementary Fig. 10). A total of 4294 movies were aligned using the patch motion correction tool and contrast

transfer function (CTF) parameters were determined by the patch CTF tool. 1600 Micrographs of estimated CTF resolution < 3.5 Å were selected for particle picking. A Topaz convolutional neural network particle picking model[73] was generated from initial blob picking and 2D classifications using 15 classes, followed by several rounds of Topaz Train. 587,742 particles were extracted in a box size of 200 by 200 pixels at a pixel size of 1.09 Å using the Topaz extract tool. After 2D classification in 100 classes, 372,284 particles (from 32 classes) were selected to generate five initial models by running the ab-initio reconstruction tool. The model corresponded to the octamer (33.9% particles) was further refined by non-uniform refinement, reaching a final resolution of 4.15 Å (GSFSC = 0.143) which was used for model building. Local-resolution and 3D-FSC plots were calculated using the local resolution tool and the "Remote 3DFSC Processing Server" web interface[74], respectively. The initial model was built based on a dimeric prediction from AlphaFold-Multimer and refined using WinCoot[69], in which the C-termini (residues 488-513) were modelled by alanine. The model was subjected to real-space refinements against the respective density maps using phenix.real_space_refine implemented in PHENIX v1.19.2[75].

The polydisperse anc$_{1b}$ datasets were processed with CryoSPARC. Particles were initially picked with the blob picker tool, extracted with a box size of 408 pixels, and subjected to iterative rounds of 2D classification. The cleaned 2D class averages were used to train a neuronal network (ResNet8) for particle picking via Topaz[73] and processed via 2D classification and the ab-initio reconstruction tool to further clean-up the data. Particles from the three datasets were first processed separately and later combined. The 2D classification job, shown in Supplementary Fig. 11c, was performed with enforced non-negativity and solvent-clamping.

### Knockout strain construction

To facilitate the characterization of CS, we deleted the native *gltA* coding region from the *E. coli* genome. To do so, we used Scarless Cas9 Assisted Recombineering (no-SCAR)[76]. We obtained plasmids from Addgene and transformed the pCas9-CR4 (ID: 62655) plasmid into *E. coli* BL21 cells. Plasmid pKDsgRNA-ack (ID: 62645) was then used as a PCR template for gRNA insertion (PCR1: GGTGTGTTCACCTTTGACCCGTTTTAGAGCTA GAAATAGCAAG; TTTATAACCTCCTTAGAGCTCGA; PCR2: GGGTCAAA GGTGAACACACCGTGCTCAGTATCTCTATCACTGA, CCAATTGTCCATA TTGCATCA). These two fragments were then combined into a single plasmid via CPEC using Q5 polymerase. This plasmid was transformed into the pCas9-CR4 containing strain. Lambda-red recombination was then induced with arabinose and the strain transformed with 5 uL of a 100 μM ssDNA fragment (T*A*A*G*TTCCGGCAGTCTTACGCAATAAGG CG-CTAAGGAGACCTTAATGATTGATTGCTAAGCCGTTTACTTTCCGGA CCCGCCTTTAATAG) that contained 45 bp of homology to the regions immediate 5' and 3' of the native *gltA* coding region. Successful deletion was confirmed via Sanger sequencing. The pKDsgRNA-ack plasmid containing the gRNA was removed by growing the resulting strain at 37 °C. The pCas9-CR4 plasmid was removed by transforming in the pKDsgRNA-p15 (ID: 62656) plasmid, which targets the pCas9-CR4 plasmid. Finally, the pKDsgRNA-p15 plasmid was removed by growing the strain at 37 °C.

### Complementation assays, Data analysis and Western Blots

The KO-strain *E.coli* BL21(DE3)ΔgltA was transformed with IPTG-inducible expression plasmids that encoded for the different CS (pLIC-backbone, same as used for protein purification). We tested growth of the transformed strains in LB-medium, where there is no selection pressure on CS activity, to test if the heterologous CS-genes had toxic effects. CS genes that impaired growth under non-selective conditions were excluded from further analysis. For growth rate experiments under CS selective conditions precultures of the complemented strains were grown overnight in LB-medium at 37 °C,

240 rpm. Cells from these overnight cultures were harvested by centrifugation at 15,900 x g for 3 min and washed two times with PBS. The washed cells were used to inoculate new cultures in 96-well plates in minimal medium (M9+glucose) to an $OD_{600}$ = 0.05. Each strain was set up in triplicates (180 μL/ well) and incubated in a microplate reader (Infinite M Nano + , Tecan, Switzerland) at 37 °C, 220 rpm for 40 h. No inducer was added during the growth experiment and CS production relied solely on leaky expression. To prevent evaporation the lid was placed onto the 96-well plates and sealed with parafilm. As a negative control we used the untransformed KO-strain *E.coli* BL21(DE3) Δ*gltA* and as positive control we complemented the KO-strain with the native CS from *E.coli* BL21(DE3) on the plasmid. Bacterial growth was monitored by absorbance measurements at 600 nm every 15 min and converted to $OD_{600}$-values via a standard curve. We used the growth rate package v0.8.2 within R studio to fit parametric growth models to the collected data and infer maximum growth rates[77].

We tested CS complementation at different protein production rates with spot assays in which the strains were grown with different concentrations of the inducer IPTG. Overnight cultures in LB were diluted in distilled water to $OD_{600}$ = 1.0. Then, a five-step serial dilution of the cell suspension was performed using a ratio of 1:5 for each step. The diluted cells were spotted on solid M9 + glucose medium with different IPTG concentrations (0, 0.01, 0.02, 0.03, 0.04, 0.05 mM) using a Singer Instruments Rotor HDA+ screening robot. Spotting was performed in a 7×7 grid with revisit of the source plate for each transfer. The plates were set up in triplicates and incubated for 72 h at 37 °C. Images of the plates were taken every 24 h using the Singer Instruments PhenoBooth + .

Protein production of the different CS homologs was analyzed via western blot. Complemented strains were grown in LB (50 mL) for 15 h, 37 °C and 240 rpm. Cells were harvested by centrifugation at 4500 x g for 20 min and resuspended in 1 mL PBS. The cells were lysed mechanically by addition of 0.1 mm glass beads and using the Fastprep24 (MP Biomedicals, USA-CA) at a strength of 6.0 for 30 s for four cycles. Cells were pelleted afterwards by centrifugation at 15,700 x g for 20 min. The supernatant was transferred and the cell pellet was resuspended in 1 mL PBS. The protein concentration of the samples was measured using a Bradford standard curve. SDS gel was loaded with 10 μg of protein for each sample and 30 μg of resuspended cell pellet. Resolved proteins were afterwards transferred onto a nitrocellulose membrane via western blotting. The membrane was blocked overnight in 5% milk, and then incubated with an Anti-histag antibody conjugated with a horseradish peroxidase for 1 hTBS-T (tris-buffered saline with tween) with 5% milk. The CS proteins were detected using chemiluminescence with a ChemiDoc imaging system (Bio-Rad, USA-CA).

### Modelling of homo-oligomeric complexes with AlphaFold2 Multimer

Structural models were generated for ancestral or extant CS using the AlphaFold2 Multimer ColabFold server[68] with default settings. Modelled structures are deposited in the source data on Edmond as described in the data availability statement. Data were rendered and visualized with PyMol (v.2.4.0).

### Reporting summary

Further information on research design is available in the Nature Portfolio Reporting Summary linked to this article.

## Data availability

Atomic structures reported in this paper are deposited to the Protein Data Bank under accession code 8QWB and 8QZP. The cryo-EM data was deposited to the Electron Microscopy Data Bank under EMD-18779. All raw data for MP spectra, kinetic traces, protein structure predictions as well as phylogenetic trees, alignments, and ancestral sequences are deposited on Edmond [https://doi.org/10.17617/3.S0HJ48], the Open Research Data Repository of the Max Planck Society for permanent public access. Source data are provided with this paper.

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

## Acknowledgements

F.L.S., T.S., G.B., D.S., and G.K.A.H. are generously supported by the Max-Planck Society. JWT acknowledges support from the NIH (R01GM131128). Cofounded by the European Union (G.K.A.H.: ERC, EVOCATION, 101040472; G.B.: ERC, KIWIsome, 101019765 ). M.G. acknowledges support by the Peter und Traudl Engelhorn Foundation.Views and opinions expressed are, however, those of the author(s) only and do not necessarily reflect those of the European Union or the European Research Council. Neither the European Union nor the granting authority can be held responsible for them. The authors acknowledge support from the EMBL Hamburg at the PETRA III storage ring (DESY, Hamburg, Germany). We are grateful for support from Sriram Garg who inferred models of oligomeric CS via Alphafold Multimer v2.

## Author contributions

F.L.S., A.P., J.W.T. and G.K.A.H. conceived the project. F.L.S and G.K.A.H. analyzed data and planned experiments. F.L.S. performed phylogenetics, enzyme kinetic measurements and analysis of the structural data. F.L.S. and T.S. inferred ancestral sequences, performed protein purification and MP measurements. A.W. and B.P.H.M. constructed the CS knockout strain. TS performed complementation assays together with D.S.; C.N.M. collected, solved, and refined the X-ray structure with supervision from G.B.; S.B., M.G., T.H. and Y.K.L. collected and processed the cryo-EM data sets. Y.K.L. refined the cryo-EM structure. JMS supervised cryo-electron microscopy. F.L.S. and G.K.A.H. wrote the manuscript with contributions and comments from all authors. C.N.M. and Y.K.L. contributed equally to this work.

## Funding

## Competing interests

The authors declare no competing interests.
