## [Transparent Peer Review file · Nature Communications]

Frequent transitions in self-assembly across the evolution of a central metabolic enzyme

Corresponding Author: Dr Georg Hochberg

Version 0:

Reviewer comments:

Reviewer #1

(Remarks to the Author)

Review of manuscript entitled "Frequent transitions in self-assembly across the evolution of a central metabolic enzyme" by Sendker, Schlotthauer et al.

In this manuscript, the authors investigate the assembly state variability of a specific protein family, namely the prokaryotic citrate synthases, across the entire phylogeny, by applying an impressive array of techniques, including mass photometry, activity assays, x-ray and cryo-EM to probe not just structure, but also the correlated enzymatic function.

As a biochemist and structural biologist, this reviewer finds this scientific question very intriguing and definitely merits deeper study. It is very often that all biochemists/structural biologists assume, as also mentioned in the introduction, that multimeric assemblies serve specific purposes, and, again, as mentioned in the introduction of the manuscript, have rationally appeared during evolution. The work presented in this manuscript challenges this assumption (to this reviewer's opinion) successfully and definitely deserves to be published in Nature Communications. Below, this reviewer lists some comments per manuscript section that should be addressed in order to increase the robustness of the manuscript:

General comments:

The authors should revisit the manuscript again and correct for italicization, typos, etc., especially in figure legends, as well as make sure that ALL figure panels are at minimum cited at the correct place in the manuscript.

Manuscript related cryo-EM maps, models should have been made available from the start to the reviewers for better assessment. Nevertheless, validation reports show no major flaws in the EM reconstructions presented in the manuscript.

Abstract

The abstract clearly communicates the background behind the work, the scientific question, and summarizes the results and research outcome quite nicely. This reviewer would suggest that the authors use the term "assembly states" instead of assemblies in line 17.

Introduction

General comments:

The introduction is short and to the point. At the end of the introduction, the authors should also include a small paragraph describing the rationale behind employing X-ray and cryo-EM both as a way to probe structure. Mass photometry should be "enough" to probe the assembly states of a multimeric enzyme, as has been previously shown by other publications. Why were only two different CS probed structurally?

Results

General comments:

The results throughout this manuscript are well described, but this reviewer finds the main figures to be a bit “too busy”. There are a lot of panels, and readability is not optimal. For example, Fig1 could be split in two (a, then b+c). Accordingly, the rest of the figures should be “thinned out”, or reconfigured, especially in cases where structural information is shown (e.g., Fig2c,d,f, Fig3e,f).

Throughout the text there are figure miss-citations, or the panels that are cited do not exist. This creates a lot of problems for the reader and should be critically addressed. In general, the figure panels in all sections, but mostly main and extended data should be reviewed again and re-organized to improve readability.

Concerning the cryo-EM structure presented in the manuscript, at the resolution reported within, side-chain placement of the residues participating in the multimeric interfaces cannot be accurately discussed and all relevant observations should be put in the context of the reported resolution.

- In the section “Hexameric assembly evolved independently in type II and III CS”, lines 14-23, an interface that is critical for multimeric assembly is discussed and characterized. The authors should also perform energetics calculations (e.g., HADDOCK) to better characterize the discussed interfaces, calculate energetics, BSA, etc. In the same section, when the results are discussed, the reader has to jump through main figure and extended data figure to get a clear picture of the described interactions. Again, figure arrangement should be re-configured to better provide the reader with the visual information required. Ext.data.fig2c,d could be moved to a new figure, along with Fig2c,d,f.
- Page 7, line 8 is here the correct supplementary figure cited? Sup.Fig.2 contains MP measurements, whereas Sup.Fig3 includes sequence alignments. If so, just from the alignments presented in Sup.Fig3, it is hard to infer the lack of the specific interfacial loop described. The authors could predict all models with AF2, align, and compare.
- Page 8, lines 1-12: Authors should clarify here that AF2-multimer-v2 was employed to predict the specific interfaces. In line 5, a panel that does not exist in the figure is cited (Fig2h), please correct. Again, energetics calculations could be done to better characterize the changes in the interface introduced by the substitutions discussed there.
- In the section “Interface turnover in type II citrate synthases” (page 8, lines 20-23), the authors state that the JK-loop is conserved in all published structures and then cite panel Fig2c which does not convey such information. All published structures should be aligned and the specific loop at the interface should be highlighted.
- Page 10, lines 1-6. Again, an interface is discussed but the cited panels (Fig3d, 1d) do not show the relevant information. In fact, again, panel Fig1d does not exist.
- Page 10, lines 17-23. An AF2-multimer prediction is an indication, not a finding. This reviewer would suggest that the language in these lines be softened and in the text it should be clarified that there is no experimental information to support this claim. Is the cited figure panel here correct? Should it not be ext.fig.3c?
- In the section “Assembly into a polydisperse distribution is an ancient trait of CS”, page 11, lines 17-20. High heterogeneity should not be an obstacle for structural characterization via cryo-EM. Latest image analysis developments allow for heterogeneity and can be tackled with 3D classification, heterogeneous refinements, etc. In the case of the results presented in Ext.data.fig5c, to corroborate hexameric assembly states, the predicted model or a published homolog should be imported in e.g., ChimeraX, converted from model to em-map at the same resolution of the presented class averages, then projected in 2D and the resulting simulated projections should be matched with the 2D classes presented here. When the word “loosely” is used, what is the context? Could a distance be measured in the 2D class averages?
- Page 13, lines 11-13: Have the authors considered that different oligomeric states could also be part of a “storage” or substrate specificity mechanism, following the paradigm of Nitrilase? In the same paragraph it is demonstrated that specific substrates shift assembly states, whereas others do not.

Discussion

General comments:

This reviewer does not find any significant issues worth mentioning in the discussion. As a suggestion, the authors could expand on the proposed experiments to probe stoichiometric variation in vivo, or in vitro. In principle, the Discussion of the manuscript is nicely written and ends with a rather controversial but thought-provoking suggestion.

Methods

General comments:

This reviewer cannot expertly judge all methodologies presented here, but after carefully reading through them, finds no significant issues. Below there are some minor comments that should be addressed.

- In the section “Kinetic enzyme assays”, page 34, lines 9-10: please add all concentrations of other substrate used to start the reaction in each case.

- In the section “Crystallography and structure determination”, please add the type of hanging-drop crystallization plates used and after how much time from plate setup crystals appeared and matured. Data completeness should also be mentioned here.
- In the section “Cryo-electron microscopy”, page 35, line 5: add the mesh of the R2/1 grids. Lines 8-9: add data collection mode (counting).
- In the section “Image processing, classification and refinement”: Cite sup.fig4. Page 36: Every time a 2D classification is mentioned, the authors should report how many classes were used to classify the particles into.

Reviewer #2

(Remarks to the Author)

The manuscript by Sender et al presents an exhaustive and detailed study of oligomerisation of citrate synthase in an evolutionary context. Remarkably, the authors show that there is no such thing as ‘functional quaternary structure’, some way essentially disproving aspects of the ever present structure-function relationship in protein function and regulation. My expertise is mostly in mass photometry, so I cannot comment on the details of the evolutionary aspects of the study, although I can say that I find the story compelling, hugely interesting and genuinely novel, and thus would consider Nature Communications to be extremely fortunate to be able to publish it (as opposed to some of its more senior siblings such as NCB, NSMB or even Nature itself). I would also suggest that the editors consider thinking about how to highlight this study (press release or similar).

I only have a few comments/thoughts for the authors to consider before publication, nothing major:

1. I sense that it may be useful to alert the general reader to the fact that ultimately ALL proteins will oligomerise, the only question is at what concentration (with the ultimate oligomerisation being crystallisation). So really what we are seeing are the consequences of relatively subtle changes in the interaction potentials, and thus equilibrium constants.
2. The real evolution that seems to have taken place here is ensuring that the active site is never obscured by oligomerisation. There seems to be something interesting here in the sense that to some degree oligomerisation is inevitable (see above) unless one goes to extremes (purely negatively charged species for example) - but at these extremes protein function if it were ever to involve protein-protein interactions (which most do) would be dead. So the trick seems to be to place the active site at an interface that will never oligomerise (extreme!), but provide others that do.
3. Combining points 1 and 2 may (or may not) be the reason why oligomerisation appears both random, but is still important. It is unavoidable (and in fact desirable) in the context of the necessity of protein-protein interactions. It just needs to happen in a way that doesn’t kill function.
4. Remind the reader that crystal structure does not equal functional protein!

Finally, out of curiosity: the widths of peaks seems to vary dramatically in a way that is not instrument limited in Fig. 1c, suggesting that some oligomeric assemblies are more polydisperse than others (e.g. compare the 334 and 462 kDa peaks at the bottom). What is happening here?

Reviewer #3

(Remarks to the Author)

In this study, the authors have explored quaternary structure variation across a large family of citrate synthases. It is admirable that the authors have taken the effort to measure homo-oligomerization across many different orthologs, as our general knowledge of how this can vary and how it is related to function is still very limited. The experiments appear to be carefully done, and I appreciate that they have complemented their mass photometry experiments with detailed structural characterizations in some cases. I think the greatest value of this work is simply in highlighting this variability. It is also interesting to discover a new type of hexameric structure within the type III family, and the eukaryotic octamer.

While overall I think this is great work, I am not totally convinced of the physiological relevance of the polydisperse quaternary structures. Although they some polydispersity does occur in the class Is, its prevalence in the ancestral reconstructions makes me suspect it is at least in part an artifact of the reconstruction. I also am fairly doubtful that these polydisperse states occur in crowded cellular environments, and I don’t think they are necessary or useful for understanding the evolution of CS quaternary structure. I don’t think it makes sense to talk about the hexamers evolving “out of a polydisperse ensemble”, which implies that contacts within the ensemble were selected for in some way - which doesn’t seem compatible with the asymmetric nature of the higher-order oligomers in the ensembles. It may be helpful for the authors to discuss these issues a little more in the paper.

Specific points:

- It took me awhile to figure out that multiple quaternary structure symbols on the tree in Fig 1 meant there were multiple species in that branch. At first I thought it was reflecting variable quaternary structure within a species. It make sense eventually, but it would be helpful to make this a bit more clear.
- It can be hard to find specific species MP plots when looking at the trees, given the way they are scattered through the text

and supplemental. Also, please include the order for the plots in Supp Fig 2.

-The paper says "we characterized the quaternary assemblies formed by 40 CS enzymes" - but there are fewer than 40 in Fig 1. Is this counting ancestral reconstructions? As written, the context implies this is referring to CS from 40 different species.

-It would be nice to have some discussion of how specific proteins chosen for MP experiments were chosen, e.g. some orders were not characterized, while in others, multiple targets were tested. Were experiments unsuccessful for some targets?

Why do some branches on the tree have no quaternary structure? Are these cases where experiments were unsuccessful?

-One of the Pseudomonadota (alpha) branches is shown as having both a dimeric and hexameric published structure. I can only find the dimer, though I didn't look too hard. Can you list all the PDBs somewhere?

-I think the complementation experiments are really interesting, and it feels like they hardly get any attention in the paper. I guess it's difficult because they are severely confounded by expression level. Is it feasible to try to quantify protein expression and compare to growth rate, to see if there are any trends that might go beyond that?

Reviewer #4

(Remarks to the Author)

The manuscript describes the analysis of the quaternary structure of citrate synthases from Archaea, Bacteria, the predicted common ancestors of the two, and the non-mitochondrial eukaryotic citrate synthases. An ongoing question in structural biology is "What is the role of different quaternary structures?" Citrate synthases are a good choice for investigating this question since they have evolved over a long period of time in all domains of life, their enzymatic activities can be measured, and different quaternary structures exist. This investigation indicates that the quaternary structure is not that important: citrate synthases have different quaternary structures that come and go as the enzyme evolves, and the quaternary structure generally has little effect on activity.

For the study, the researchers used phylogenetics to construct the evolutionary pathway and ancestral sequence reconstruction to predict common ancestors. The tree was used to select citrate synthases for experimental investigation. Experimental approaches to investigate the quaternary structure included mass photometry, X-ray crystallography, and cryoelectron microscopy. The experimental data, statistical analysis, and conclusions are appropriate. I have little experience with phylogenetics or ancestral sequence reconstruction so cannot say how reliable the evolutionary tree or the sequences of the proteins at the branching points (i.e., the ancestral proteins) are.

The research is significant because it answers an ongoing question in structural biology using the citrate synthases as the test case. The authors cite two other analyses that have recently been published using geranylgeranylgeranyl glyceryl phosphate synthase and Rubisco and describe their own work as an extension.

A limitation of the research is that the analysis is predominantly *in vitro*. The *in vivo* experiments using knockout of *E. coli* *gltA* (the gene for the native citrate synthase, which is an allosterically regulated hexamer) and inclusion of a plasmid expressing the gene for a different citrate synthase shows that the different citrate synthase can complement the knockout, at least under the lab conditions tested. This showed that the hexameric quaternary structure was not essential for life. Experiments with the non-mitochondrial eukaryotic citrate synthase indicated a low level of protein production, but the question of whether different quaternary structures lead to different protein stability is not addressed. As well, the quaternary structure is measured *ex vivo*, without other proteins with which citrate synthase might interact. Such experiments are outside the scope of this work.

Supplementary Figure 1 is not legible. Otherwise, the manuscript is clear.

Reviewer #5

(Remarks to the Author)

This paper touches on an interesting question in the evolution of proteins, i.e., if the oligomerization of proteins has evolutionary meaning or not. The question is certainly underexplored and there are no systematic studies that allowed us to discuss what are driving forces for altering oligomerization during evolution. The work presents a relatively large-scale analysis of the oligomerization of a protein family (CS) and shows a comprehensive picture of how oligomerizations alter during protein divergence. The comprehensive experimental data for oligomerization is certainly a great contribution of this work and the claim that oligomerization can be evolved randomly with no strong relation to the functionality of proteins, thus rather neutral events during evolution. The overall conclusion derived from the work is sensible and supports a good amount of experimental data, while it is not surprising to me at all. I appreciate the good quality and amount of the data that the authors provided allowed, for the first time, to discuss potential driving forces for the oligomerizations. While I have one major reservation for the paper (and a few experimental suggestions), but I believe that the work has a substantial interest to the readers of Nature Communications.

As a biochemist, who purified many protein variants in my career, the conclusion that the draw is not really surprising at all. It is not uncommon to observe that a mutation or two alter the oligomerization state of proteins, in particular, many variants exhibit polydisperse phenotypes in my experience. So, I naturally would think that many oligomerizations can be just neutral events as far as the protein can be expressed stably in the cell and function. While I recognize that, in some cases, oligomerization occurs for evolutionary advantage (stability, allostery etc), I believe that most, if not all, protein scientists would respond the same. Thus, I felt that Introduction sounds very rough and one-sided to state that the consensus in the

community is that oligomerization always comes with adaptation to selection pressures. I acknowledge that previous published papers and reviews often claimed that oligomerization has evolutionary meanings.

Nonetheless, it would be appropriate to give a more balanced introduction in the abstract, introduction and discussion.

More importantly, I found that the claim that ancestral CS are polydisperse clumsy and it must come with much more careful discussion. As I stated above, observing polydisperse oligomers is not uncommon in protein variants, one or two mutations can easily alter oligomeric states from my knowledge. As the authors stated, the observed phenotypes of ancestral CSs can be coming from the inaccuracy of ancestral sequences by predictions. When we are analyzing something relatively robust property of proteins, e.g. protein function, the predicted ancestral trait might be relatively trustable. One must be extremely careful when we discuss the ancestral state of more sensitive and changeable traits, e.g., protein stability and oligomeric states can be easily altered by any mutations across the entire proteins.

I am not sure if observing polydisperse extant enzymes can be a sufficient support that the observed predicted ancestral CSs' phenotype is true. Do the authors imply that polydispersity is a "acceptable" form in living organisms? Probably, there are many other possibilities why polydisperse forms can be observed in the authors system, e.g., polydisperse forms can be caused by the lack of endogenous chaperones of the original host. Also, only way to address the inaccuracy of ancestral reconstruction is to explore the sequence space around predicted ancestral states (including combinations) and see if monodisperse variants exist or not, and how mutations alter polydisperse phenotypes. Currently, there is no efforts to address this issue in the manuscript. Unless the authors provide additional and rather extensive convincing experimental data, I strongly believe that the claim should be torn down substantially and discuss with a caveat of high probability of prediction bias and inaccuracy of ancestral reconstruction.

Related to polydispersity, I wonder how protein stability and solubility are altered and related to oligomerization? There simple protein traits are sometime related oligomerization, and I am surprised that the authors did not measure any of those properties when they purified all proteins and tested expression in *E. coli*. The authors showed western blots of CSs in Extended Data Fig.7, but there is no measure of soluble and insoluble expression (at least soluble and insoluble expression with overexpression for protein purification). I believe that such analysis can inform some properties that are related to oligomerization, e.g., aggregation propensity of CSs (thus polydispersity), especially predicted ancestral CSs? Also thermostability or other in vitro stability measurements can be done to estimate the effect of oligomerizations and polydispersity. I personally would love to see the data in this manuscript (as it will make the paper more interesting and informative), but would not like to request it firmly, rather would like to let the authors decide to perform and include them in the revised paper, or consider it as a future work.

Version 1:

Reviewer comments:

Reviewer #1

(Remarks to the Author)

Comments to the authors (Sendker, Schlotthauer et al., Frequent transitions in self-assembly across the evolution of a central metabolic enzyme):

The authors of the manuscript did what I believe is an amazing job addressing all my concerns during the first round of revisions. Their answers to my comments were well-reasoned, exhaustive, and I feel that the additions/alterations made to the manuscript now further elevate its already high quality. To this end, I recommend this manuscript for publication, without any further comments that need to be addressed. Again, I congratulate the authors for their thought-provoking and robust work, that truly contributes to the field.

Reviewer #2

(Remarks to the Author)

I have no further comments and am happy to recommend publication

Reviewer #3

(Remarks to the Author)

I am happy with the way that the authors have addressed all of my comments.

Reviewer #4

(Remarks to the Author)

I am satisfied with the revisions made by the authors.

Point by point response:

Reviewer 1

- *The authors should revisit the manuscript again and correct for italicization, typos, etc., especially in figure legends, as well as make sure that ALL figure panels are at minimum cited at the correct place in the manuscript.*

A: We thank the reviewer for careful reading of the document and have worked through the manuscript to correct for typos and figure legend errors.

- *The abstract clearly communicates the background behind the work, the scientific question, and summarizes the results and research outcome quite nicely. This reviewer would suggest that the authors use the term “assembly states” instead of assemblies in line 17.*

A: Agreed and changed accordingly (l. 37).

- *The introduction is short and to the point. At the end of the introduction, the authors should also include a small paragraph describing the rationale behind employing X-ray and cryo-EM both as a way to probe structure. Mass photometry should be “enough” to probe the assembly states of a multimeric enzyme, as has been previously shown by other publications. Why were only two different CS probed structurally?*

A: We focused on “only” these two as there are deposited structures for other types of CS enzymes already. We investigated the structure of CS enzymes that we have found to be evolutionarily novel meaning they have not reported before for citrate synthases. These are the type III hexamer and the octamer found in eukaryotic non-mitochondrial CS enzymes. Solving more structures would certainly reveal interesting information about the turnover at interface sites, but ultimately was beyond the scope of this study.

We added this rationale briefly to the introduction (l. 79)

- *The results throughout this manuscript are well described, but this reviewer finds the main figures to be a bit “too busy”. There are a lot of panels, and readability is not optimal. For example, Fig1 could be split in two (a, then b+c). Accordingly, the rest of the figures should be “thinned out”, or reconfigured, especially in cases where structural information is shown (e.g., Fig2c,d,f, Fig3e,f).*

A: We recognize the reviewer’s concern. Since we have only 5 main display items available, we kept the number of main figures as is. But we reorganized the panels to make them less busy.

- *Throughout the text there are figure miss-citations, or the panels that are cited do not exist. This creates a lot of problems for the reader and should be critically addressed. In general, the figure panels in all sections, but mostly main and extended data should be reviewed again and re-organized to improve readability.*

A: We appreciate the careful reading and have addressed this issue and corrected the figure miss-citations and re-organized part of the figures to improve readability (see above).

- *Concerning the cryo-EM structure presented in the manuscript, at the resolution reported within, side-chain placement of the residues participating in the multimeric interfaces cannot be accurately discussed and all relevant observations should be put in the context of the reported resolution.*

A: We do not discuss side-chain placements in the manuscript of the reported cryo-EM structure. The observations we made – the interaction between subunits to form octamers is conferred by the elongated C-terminus – is based purely on the EM-density shown in Fig. 3f and was additionally verified by mutational studies in which the respective amino acids were deleted from the protein (Fig. 3g).

We have highlighted now the restrictions of the resolution additionally in the text (ll. 294-296).

- *In the section “Hexameric assembly evolved independently in type II and III CS”, lines 14-23, an interface that is critical for multimeric assembly is discussed and characterized. The authors should also perform energetics calculations (e.g., HADDOCK) to better characterize the discussed interfaces, calculate energetics, BSA, etc.*

A: This is a thoughtful suggestion but we ultimately decided against this. Our paper directly quantifies the contributions of individual substitutions by measuring their effects empirically. Computational approaches (as the reviewer suggests) would certainly add more fine grained information, but we simply felt that this was beyond the scope of this already very substantial study.

- *In the same section, when the results are discussed, the reader has to jump through main figure and extended data figure to get a clear picture of the described interactions. Again, figure arrangement should be re-configured to better provide the reader with the visual information required. Ext.data.fig2c,d could be moved to a new figure, along with Fig2c,d,f.*

A: We have rearranged the mentioned figures accordingly.

- *Page 7, line 8 is here the correct supplementary figure cited? Sup.Fig.2 contains MP measurements, whereas Sup.Fig3 includes sequence alignments. If so, just from the alignments presented in Sup.Fig3, it is hard to infer the lack of the specific interfacial loop described. The authors could predict all models with AF2, align, and compare.*

A: Yes, there was a mistake with the cited supplementary figure. The correct one is Supplementary figure 3 (now Supp. Fig. 4). Panel a shows a sequence alignment of the inferred ancestors which shows on the sequence level that the residues of the interfacial loop are not present. We have now highlighted the respective gaps in the figure to make it more obvious. We also predicted the structures of the ancestral sequences with AF2 to also show structurally that the loop is not present and added them to the Figure (Supplementary Fig. 4b-c)

- Page 8, lines 1-12: Authors should clarify here that AF2-multimer-v2 was employed to predict the specific interfaces. In line 5, a panel that does not exist in the figure is cited (Fig2h), please correct. Again, energetics calculations could be done to better characterize the changes in the interface introduced by the substitutions discussed there.

A: We have now clarified that AF2-multimer was employed to predict the specific interfaces (l. 203-204). We also have corrected the figure cited to the panel (Fig. 2f.). We refer the reviewer to our answer above about energy calculations.

- *In the section “Interface turnover in type II citrate synthases” (page 8, lines 20-23), the authors state that the JK-loop is conserved in all published structures and then cite panel Fig2c which does not convey such information. All published structures should be aligned and the specific loop at the interface should be highlighted.*

A: We have added a panel to the figure that shows a structural alignment of all published type II structures and their respective JK-loops (Ext. Data Fig. 3a).

(a) Structural alignment of deposited type II CS, all have the JK-loop (PDB: 4E6Y, 6ZU0, 2H12, 4G6B, 4XGH, 7E8N, 4TVM, 3MSU).

- *Page 10, lines 1-6. Again, an interface is discussed but the cited panels (Fig3d, 1d) do not show the relevant information. In fact, again, panel Fig1d does not exist.*

A: We discuss an *inference* not an interface but indeed one of the panels is wrong. We have corrected the cited panel to Fig. 1c (l. 293).

- *Page 10, lines 17-23. An AF2-multimer prediction is an indication, not a finding. This reviewer would suggest that the language in these lines be softened and in the text it should be clarified that there is no experimental information to support this claim. Is the cited figure panel here correct? Should it not be ext.fig.3c?*

A: We have softened the language as requested (ll. 294-306).

The cited figure panel shows the predicted involvement of the C-terminal extension in the interaction between subunits. But Ext. Figure 3c shows the predicted JK-loop

that makes the typical connection within the hexamer. It is probably best to cite both panels here. We have adapted it accordingly (now Ext. Data Fig. 3d-e).

- *In the section “Assembly into a polydisperse distribution is an ancient trait of CS”, page 11, lines 17-20. High heterogeneity should not be an obstacle for structural characterization via cryo-EM. Latest image analysis developments allow for heterogeneity and can be tackled with 3D classification, heterogeneous refinements, etc.*

A: We agree with the reviewer that image analysis developments have greatly improved over the last few years. However, structural heterogeneity often still is a major bottleneck in obtaining high-resolution structural insights and the success of tackling heterogeneity via image processing techniques varies from case to case. The polydisperse anc_{1a} sample is inherently structurally heterogenous and the 2D class averages, shown in Extended Data Fig. 5, were obtained via extensive cleaning via iterative rounds of 2D classification and ab-initio reconstructions. This procedure reduced the number of particles from 817,849 (from three datasets) down to 44,251 particles. Subjecting these particles to 3D non-uniform refinement in CryoSPARC yielded a 3D reconstruction at a nominal resolution of 8 Å (see Fig. R1a, below). Further classification did not improve the quality of the map, presumably due to further reduction of particles. Nevertheless, with three density blobs, the 3D map still implies a trimeric/hexameric (trimer of CS dimers) stoichiometry. Both the 2D classes as well as the 3D map (see arrows in Fig. R1a) furthermore indicate that the CS forms non-symmetric higher-order oligomers, which is already informative. The reviewer might, however, well agree that the given resolution of 8 Å presumably is an overestimation as no secondary structure elements are visible. We, therefore, prefer a conservative interpretation of the EM results and therefore would like to only show the 2D class averages in the manuscript.

- *In the case of the results presented in Ext.data.fig5c, to corroborate hexameric assembly states, the predicted model or a published homolog should be imported in e.g., ChimeraX, converted from model to em-map at the same resolution of the presented class averages, then projected in 2D and the resulting simulated projections should be matched with the 2D classes presented here.*

A: We followed the reviewer’s request of comparing our 2D class average with a symmetric hexameric CS assembly and reprocessed previously collected dataset of a hexameric CS from *S. elongatus* (Δ 2-6 SeCS, PDB: 8BEI, EMD-16004 Ref: Sendker et al. Nature 628, 894–900 (2024). <https://doi.org/10.1038/s41586-024-07287-2>). To better compare the 2D class averages of both samples, we low-pass filtered the particle images to a resolution of 15 Å, limited the dataset to 20,000 particles and subjected the particles to 2D classification. To directly compare

selected 2D class averages side-by-side, we selected representative views of both datasets (resulting in a total of ca. 9,000 particles for $\Delta 2-6$ SeCS and ca. 20,000 particles of anc_{1a}) (Fig. R1b, below). Notably, the hexameric $\Delta 2-6$ SeCS structure adopts a barrel-like structure, and the class-averages mostly represent side-views onto the barrel, which makes a direct comparison difficult. Nevertheless, top-views, illustrating the hexameric/trimeric nature of CS dimers, are also present (see red frames in Fig. R1b). This view is comparable to some of our anc_{1a} class averages. Yet, this comparison also indicates that the $\Delta 2-6$ SeCS is of symmetric nature whereas the anc_{1a} hexamers are non-symmetric as the densities, corresponding to CS dimers appear to be unevenly oriented. We, again, think that the side-by-side comparison is not strictly necessary to show that the anc_{1a} hexamers are of a different (namely non-symmetric) nature compared to the $\Delta 2-6$ SeCS hexamers, and, therefore, would prefer to not implement this analysis into the final manuscript to aid readability.

Figure R1. Additional analysis, prepared for the reviewers as part of the point-by-point response to illustrate the non-symmetric nature of the polydisperse anc_{1a} sample. **a.** Non-uniform refinement of extensively cleaned CS particles (successive round of 2D classification and ab-initio reconstructions), calculate with CryoSPARC. **b.** Comparison between reprocessed Δ 2-6 SeCS (Sendker et al., EMD-16004) and anc1a hexamer 2D class averages. The Δ 2-6 SeCS particles were lowpass filtered as indicated to aid the direct comparison.

- *When the word “loosely” is used, what is the context? Could a distance be measured in the 2D class averages?*

A: We agree that the term “loosely” can be confusing and, therefore, removed it in the revised text.

- *Page 13, lines 11-13: Have the authors considered that different oligomeric states could also be part of a “storage” or substrate specificity mechanism, following the paradigm of Nitrilase? In the same paragraph it is demonstrated that specific substrates shift assembly states, whereas others do not.*

A: Interesting question. The change of oligomeric state as a mechanism that influences substrate specificity seems not very likely in this case as we do not observe differences in K_m for the native substrate in different oligomeric states for this CS. A storage form that is connected to oligomeric assembly could be a possibility as this has been observed in other enzymes and we have added to the manuscript (l. 407-410).

- *In the section “Kinetic enzyme assays”, page 34, lines 9-10: please add all concentrations of other substrate used to start the reaction in each case.*

A: The information was added to the manuscript (ll 962-964).

- *In the section “Crystallography and structure determination”, please add the type of hanging-drop crystallization plates used and after how much time from plate setup crystals appeared and matured. Data completeness should also be mentioned here.*

A: We have added the requested additional information to the method section “Crystallography and structure determination” (ll. 984-991).

- *In the section “Cryo-electron microscopy”, page 35, line 5: add the mesh of the R2/1 grids. Lines 8-9: add data collection mode (counting).*

A: We have added the requested information to the methods part (l. 1005; l. 1009)

- *In the section “Image processing, classification and refinement”: Cite sup.fig4.*

A: The citation was added (l. 1021)

- *Page 36: Every time a 2D classification is mentioned, the authors should report how many classes were used to classify the particles into.*

A: We have added the number of classes that were used for classification at each step (ll. 1025-1027)

Reviewer 2

We thank the reviewer for his enthusiastic assessment and the interesting comments.

- *1. I sense that it may be useful to alert the general reader to the fact that ultimately ALL proteins will oligomerise, the only question is at what concentration (with the ultimate oligomerisation being crystallisation). So really what we are seeing are the consequences of relatively subtle changes in the interaction potentials, and thus equilibrium constants.*

2. The real evolution that seems to have taken place here is ensuring that the active site is never obscured by oligomerisation. There seems to be something interesting here in the sense that to some degree oligomerisation is inevitable (see above) unless one goes to extremes (purely negatively charged species for example) - but at these extremes protein function if it were ever to involve protein-protein interactions (which most do) would be dead. So the trick seems to be to place the active site at an interface that will never oligomerise (extreme!), but provide others that do.

3. Combining points 1 and 2 may (or may not) be the reason why oligomerisation appears both random, but is still important. It is unavoidable (and in fact desirable) in the context of the necessity of protein-protein interactions. It just needs to happen in a way that doesn't kill function.

A: We agree with the reviewer. We mention the point that oligomerization apparently occurs in a way that preserves function in the discussion. The reviewer is right that oligomerization is perhaps inevitable at some concentration, but it is by no means obvious that we would expect it to be inevitable at very low (intracellular)

concentrations. Nor is it obvious to us that assembly into closed symmetries is inevitable at high concentrations before aggregation or crystallization occurs. Though we agree with the reviewer in principle, we therefore felt that such a statement would require too many qualifiers for a short discussion, but we do plan to address this in an upcoming review.

- *4. Remind the reader that crystal structure does not equal functional protein!*

A: Agreed and added to the introduction (l. 61-62).

- *Finally, out of curiosity: the widths of peaks seems to vary dramatically in a way that is not instrument limited in Fig. 1c, suggesting that some oligomeric assemblies are more polydisperse than others (e.g. compare the 334 and 462 kDa peaks at the bottom). What is happening here?*

A: That is a good observation! But the reason is indeed instrument limited. In the pointed to example the measurement with the 334 kDa peak (P. olseni) was measured on an OneMP mass photometer and the sample with the 462 kDa peak (A. comosus) was measured on a TwoMP instrument which has better sensitivity and results in narrower peaks.

Reviewer 3

While overall I think this is great work, I am not totally convinced of the physiological relevance of the polydisperse quaternary structures. Although they some polydispersity does occur in the class Is, its prevalence in the ancestral reconstructions makes me suspect it is at least in part an artifact of the reconstruction. I also am fairly doubtful that these polydisperse states occur in crowded cellular environments, and I don't think they are necessary or useful for understanding the evolution of CS quaternary structure. I don't think it makes sense to talk about the hexamers evolving "out of a polydisperse ensemble", which implies that contacts within the ensemble were selected for in some way - which doesn't seem compatible with the asymmetric nature of the higher-order oligomers in the ensembles.

It may be helpful for the authors to discuss these issues a little more in the paper.

A: We thank the reviewer for the thoughtful and productive comments. Regarding the remarks on polydispersity and its prevalence in the ancestral proteins of type I CS: To test the robustness of this phenotype we have now resurrected also alternative ancestral proteins that contain the second most likely state for positions that are more uncertain in the sequence (PP>0.2). These proteins differ in 56-69 positions from the maximum-likelihood ancestors but still retain the ability, albeit slightly weaker, to assemble into polydisperse distributions (Supplementary Fig. 6). This, together with the three evolutionary

distant CS enzymes that also retain this behavior, are convincing results to us that this phenotype is not simply an artefact of the method.

Regarding physiological relevance, we actually make a point in the manuscript that this assembly might in fact not be physiologically relevant or visible to natural selection (l 357-359; 480-481). The individual complexes remain catalytically competent enzymes and their abundance is not very high compared to the dimeric subcomplexes. Many of the polydisperse enzymes also complement function in *E. coli* very well (Fig. 5g) and we therefore deem this phenotype mostly to be a form of harmless complexity that can be encoded by many functional CS sequences.

Regarding the evolution of hexamers from these polydisperse ensembles: the reviewer is probably right: The hexamers we see in the polydisperse ensemble are most likely structurally distinct from monodisperse hexamers. We have softened our language here.

- *It took me awhile to figure out that multiple quaternary structure symbols on the tree in Fig 1 meant there were multiple species in that branch. At first I thought it was reflecting variable quaternary structure within a species. It make sense eventually, but it would be helpful to make this a bit more clear.*

A: We understand the confusion and have added additional information to the figure and its legend.

- *It can be hard to find specific species MP plots when looking at the trees, given the way they are scattered through the text and supplemental. Also, please include the order for the plots in Supp Fig 2.*

A: We have now added a schematic phylogenetic tree to the figure, ordered the MP spectra accordingly and mapped them onto the tree (Supplementary Fig. 2).

- The paper says "we characterized the quaternary assemblies formed by 40 CS enzymes" - but there are fewer than 40 in Fig 1. Is this counting ancestral reconstructions? As written, the context implies this is referring to CS from 40 different species.

A: With 40 CS enzymes we mean 40 extant CS homologs from different species. The reason why there are not 40 symbols on the tree in Fig. 1 is readability. We only added a symbol to represent the different quaternary structures that have been found for each clade. If we characterized two that assemble into the same quaternary structure per clade, we did not represent this with two symbols. If you count all the MP spectra in all figures, they come up to 40 extant ones. The forty characterized species are also shown in the full tree in Supplementary Fig. 1 (but it has been brought to our attention that this figure was not legible for some reviewers for unknown reasons and we apologize for that).

- *It would be nice to have some discussion of how specific proteins chosen for MP experiments were chosen, e.g. some orders were not characterized, while in others, multiple targets were tested. Were experiments unsuccessful for some targets? Why do some branches on the tree have no quaternary structure? Are these cases where experiments were unsuccessful?*

A: There was no particularly specific strategy in characterizing the different CS homologs from the beginning. In general, we tried to characterize at least one enzyme from each phylogenetic clade shown for the tree in Fig. 1. And yes, some of the experiments were unsuccessful in the sense that we could not obtain soluble protein and therefore were not able to assign a quaternary structure. For other clades we characterized multiple CS mostly because we had found an unexpected assembly state in this clade. Here we wanted to investigate if this is a single anomaly or if this is found in multiple species in that clade to also get an idea when it evolved. We adapted the introduction to reflect this better (ll 123-127).

- *One of the Pseudomonadota (alpha) branches is shown as having both a dimeric and hexameric published structure. I can only find the dimer, though I didn't look too hard. Can you list all the PDBs somewhere?*

The hexameric CS of the Pseudomonadota (alpha clade) is *Acetobacter aceti* (PDB 2H12). We have now added all PDB accession codes to Supplementary Fig. 2b.

- *I think the complementation experiments are really interesting, and it feels like they hardly get any attention in the paper. I guess it's difficult because they are severely confounded by expression level. Is it feasible to try to quantify protein expression and compare to growth rate, to see if there are any trends that might go beyond that?*

A: The reviewer's comment is correct. We have no good way to control for protein expression levels, so our conclusions have to remain limited here. We used Western blots to quantify protein production, but comparison with growth rates was

not meaningful because many proteins were not expressed. For all other proteins, growth rate was found to correlate with protein production.

Reviewer 4

- *Supplementary Figure 1 is not legible. Otherwise, the manuscript is clear.*

A: We apologize that something went wrong with Supplementary Fig. 1 and it was not legible. It is not clear to us how that happened, but we hope it is solved now.

→ Additional comments/formality-related recommendations the reviewer made in the manuscript and that were sent in a separate PDF file have been adapted in the manuscript.

Reviewer 5

- *As a biochemist, who purified many protein variants in my career, the conclusion that the draw is not really surprising at all. It is not uncommon to observe that a mutation or two alter the oligomerization state of proteins, in particular, many variants exhibit polydisperse phenotypes in my experience. So, I naturally would think that many oligomerizations can be just neutral events as far as the protein can be expressed stably in the cell and function. While I recognize that, in some cases, oligomerization occurs for evolutionary advantage (stability, allostery etc), I believe that most, if not all, protein scientists would respond the same. Thus, I felt that Introduction sounds very rough and one-sided to state that the consensus in the community is that oligomerization always comes with adaptation to selection pressures. I acknowledge that previous published papers and reviews often claimed that oligomerization has evolutionary meanings. Nonetheless, it would be appropriate to give a more balanced introduction in the abstract, introduction and discussion.*

A: We appreciate the reviewer's openness to the idea that changes in oligomerization can occur neutrally in evolution. But we do not agree that this is a consensus among biochemists. As the reviewer acknowledges himself, many previously published papers indicate homo-oligomeric assembly with some form beneficial function for the organism. We therefore do not agree that our introduction is too one-sided. We introduce the commonly associated benefits of homo-oligomeric protein assembly and point out that there are only few studies that directly show correlation and discuss the previous limitations in assigning quaternary structures accurately and in higher throughput. The feedback of the other reviewers is also consistent with our view that the connection between adaptive benefit and oligomeric assembly is limited and that the structure-function relationship is still the predominant way to think about proteins. Our introduction

also cites dissenting opinions, so overall we feel our writing presents the issue as undecided.

- *More importantly, I found that the claim that ancestral CS are polydisperse clumsy and it must come with much more careful discussion. As I stated above, observing polydisperse oligomers is not uncommon in protein variants, one or two mutations can easily alter oligomeric states from my knowledge. As the authors stated, the observed phenotypes of ancestral CSs can be coming from the inaccuracy of ancestral sequences by predictions. When we are analyzing something relatively robust property of proteins, e.g. protein function, the predicted ancestral trait might be relatively trustable. One must be extremely careful when we discuss the ancestral state of more sensitive and changeable traits, e.g., protein stability and oligomeric states can be easily altered by any mutations across the entire proteins.*

A: We find the reviewers comment that polydisperse oligomers are not uncommon in protein variants a little surprising as we are not aware of many proteins which have been described as polydisperse before (only sHSPs and α -Crystallin come to mind). We do think that this rarity is due to a discovery bias and previous techniques (except native mass spectrometry) lack the resolution to actually detect them. But it is still regarded as a rather uncommon protein property until now. In combination with the other comments made by the reviewer in which polydispersity and aggregation-propensity are equated (see below), we think there might be some confusion or a difference in how the term “polydispersity” is used.

The way we use “polydispersity” as a phenotype of proteins in this article (and which has been employed by other authors that investigated this phenomenon with e.g. native mass spectrometry) is the simultaneous assembly into multiple quaternary structures that are distinct and stable. We show that this is the case for CS enzymes by separating them with size exclusion chromatography and showing that the different complexes retain stability and catalytic activity. This is different to random oligomerization or aggregation into larger complexes that can appear due to protein folding impairment which is known to be inducible by “one or two” mutations and we think the reviewer is referring to.

In addition, there is a difference between knowing that oligomeric state can be altered in a few mutations (which has been demonstrated before, including by us) and showing that such mutations actually fix in natural proteins. The main contribution of our study is to show that oligomeric state really does vary substantially across a protein family with a very conserved function.

- *I am not sure if observing polydisperse extant enzymes can be a sufficient support that the observed predicted ancestral CSs' phenotype is true. Do the authors imply that polydispersity is a "acceptable" form in living organisms? Probably, there are many other possibilities why polydisperse forms can be observed in the authors*

system, e.g., polydisperse forms can be caused by the lack of endogenous chaperones of the original host.

A: With the data presented in the manuscript, we do not see a reason why polydispersity should be an “unacceptable” form of assembly for a living organism: We show that the different assembly states are stable and do not keep growing into larger aggregates (Extended Data Fig. 5a). We show that all stoichiometries are catalytically competent enzymes (Fig. 5a-b). And we show that polydisperse enzymes can complement function in *E. coli* and not only rescue survival but allow growth that is comparable to that of its native CS enzyme (Fig. 5g).

In addition, we are not aware of a citrate synthase that is dependent on a chaperone but if that is the case for one of the polydisperse enzymes we doubt that *E. coli* could reconstitute the specific function from its original host and therefore explain the complementation.

We also do not make strong claims about the physiological relevance of polydisperse assembly of CS enzymes. Instead, we even argue that polydispersity might have no or only limited influence on the enzyme’s function and therefore represents a form of harm- or useless complexity. But since we find this propensity in three independent extant CS enzymes besides the reconstructed ancestors, we do not agree that this should be taken as an artifact of the system.

Even if the reviewer views our reconstructions as artefactual sequences, they at least prove that function-preserving changes on a protein’s surface can lead to this kind of assembly without causing substantial harm. This alone implies that this kind of behavior has no reason not to evolve in enzymes. We see no reason to exclude it as an unacceptable kind of oligomerization, purely based on its lack of symmetry.

- *Also, only way to address the inaccuracy of ancestral reconstruction is to explore the sequence space around predicted ancestral states (including combinations) and see if monodisperse variants exist or not, and how mutations alter polydisperse phenotypes. Currently, there is no effort to address this issue in the manuscript. Unless the authors provide additional and rather extensive convincing experimental data, I strongly believe that the claim should be torn down substantially and discuss with a caveat of high probability of prediction bias and inaccuracy of ancestral reconstruction.*

A: We agree that alternative ancestral reconstructions that explore the sequence space around the predicted ancestral states are a good way to inform about the robustness of the inferred trait. Therefore, we have now inferred and purified alternative ancestral CS proteins that contain less likely amino acid residues at positions that were more uncertain (PP > 0.2).

These alternative ancestors of the polydisperse type I CS differ in 56-69 residues from the maximum likelihood ancestors but retain the polydisperse phenotype

(albeit slightly weaker). This convinces us that this phenotype is very robust and can be encoded by many CS sequences. Together with the results from three distantly related, extant CS enzymes that show polydispersity we do not think this merits our claim to be “torn down substantially”.

- *Related to polydispersity, I wonder how protein stability and solubility are altered and related to oligomerization? There simple protein traits are sometime related oligomerization, and I am surprised that the authors did not measure any of those properties when they purified all proteins and tested expression in E. coli. The authors showed western blots of CSs in Extended Data Fig.7, but there is no measure of soluble and insoluble expression (at least soluble and insoluble expression with overexpression for protein purification). I believe that such analysis can inform some properties that are related to oligomerization, e.g., aggregation propensity of CSs (thus polydispersity), especially predicted ancestral CSs?*

A: We did perform western blots of the soluble and insoluble protein production in *E. coli* but the insoluble fractions were not added to the manuscript as they did not provide a lot of additional information. We have added them now to Extended Data Fig. 7 (see below). There is only one example in which a protein is observed to be clearly more abundant in the insoluble fraction (*Myxococcota sp.*) which is one that is found to assemble into dimers. The western blots show no indication that proteins that assemble into a polydisperse distribution are more aggregation-prone than other oligomeric assemblies. As we discussed in the comment above, we do not equal polydispersity with aggregation propensity! We have highlighted the polydisperse proteins in the figure in this document:

- Also, thermostability or other *in vitro* stability measurements can be done to estimate the effect of oligomerizations and polydispersity. I personally would love to see the data in this manuscript (as it will make the paper more interesting and informative), but would not like to request it firmly, rather would like to let the authors decide to perform and include them in the revised paper, or consider it as a future work.

A: We thank the reviewer for this recommendation. We agree that thermostability is an interesting additional aspect of oligomerization. We have investigated this for the evolution of Type III hexamers and have found that the introducing the interface responsible for this seems to have a negligible effect on stability. Abolishing it in a type III hexamers conversely appears to destabilize the protein, but not enough to affect catalysis at physiological temperatures (lines 438-495, Extended Data Fig. 6c, e). This implies that the interface became somewhat entrenched after its initial gain, but not enough to completely prevent a future loss. This may in turn explain why interfaces between dimers are relatively frequently lost in our phylogeny.